# A developmental biliary lineage program cooperates with Wnt activation to promote cell proliferation in hepatoblastoma

Peng V. Wu [1,2,3,4,8,9] ✉, Matt Fish[1,2,3], Florette K. Hazard[5,10], Chunfang Zhu[5], Sujay Vennam[5], Hannah Walton[1,2,3,11], Dhananjay Wagh[6], John Coller[6], Joanna Przybyl[5,12,13], Maurizio Morri[7,14], Norma Neff [7], Robert B. West[5] & Roel Nusse [1,2,3] ✉

Cancers evolve not only through the acquisition and clonal transmission of somatic mutations but also by epigenetic mechanisms that modify cell phenotype. Here, we use histology-guided and spatial transcriptomics to characterize hepatoblastoma, a childhood liver cancer that exhibits significant histologic and proliferative heterogeneity despite clonal activating mutations in the Wnt/β-catenin pathway. Highly proliferative regions with embryonal histology show high expression of Wnt target genes, the embryonic biliary transcription factor *SOX4*, and striking focal expression of the growth factor *FGF19*. In patient-derived tumoroids with constitutive Wnt activation, FGF19 is a required growth signal for FGF19-negative cells. Indeed, some tumoroids contain subsets of cells that endogenously express *FGF19*, downstream of Wnt/β-catenin and SOX4. Thus, the embryonic biliary lineage program cooperates with stabilized nuclear β-catenin, inducing FGF19 as a paracrine growth signal that promotes tumor cell proliferation, together with active Wnt signaling. In this pediatric cancer presumed to originate from a multipotent hepatobiliary progenitor, lineage-driven heterogeneity results in a functional growth advantage, a non-genetic mechanism whereby developmental lineage programs influence tumor evolution.

Cancers are classically thought to develop through clonal evolution, whereby sequential somatic mutations give rise to heterogeneous cell lineages subject to selective forces, leading to the emergence of subclones[1,2]. On the other hand, epigenetic mechanisms, which generate the vast heterogeneity of cells and tissues arising during normal embryogenesis, can significantly expand the diversity of cells carrying the same genetic mutations[3–5]. Such paths toward intratumoral heterogeneity may be particularly relevant for pediatric

[1]Howard Hughes Medical Institute, Stanford University School of Medicine, Stanford, CA 94305, USA. [2]Department of Developmental Biology, Stanford University School of Medicine, Stanford, CA 94305, USA. [3]Institute for Stem Cell Biology and Regenerative Medicine, Stanford University School of Medicine, Stanford, CA 94305, USA. [4]Department of Pediatrics, Stanford University School of Medicine, Stanford, CA 94305, USA. [5]Department of Pathology, Stanford University School of Medicine, Stanford, CA 94305, USA. [6]Stanford Genomics, Stanford University, Stanford, CA 94305, USA. [7]Chan Zuckerberg Biohub, Stanford, CA 94305, USA. [8]Present address: Division of Oncology, Cincinnati Children's Hospital Medical Center, Cincinnati, OH 45229, USA. [9]Present address: Department of Pediatrics, University of Cincinnati College of Medicine, Cincinnati, OH 45229, USA. [10]Present address: Department of Pathology and Laboratory Medicine, University of California Davis School of Medicine, Sacramento, CA 95817, USA. [11]Present address: Department of Population Health, NYC Health + Hospitals, New York, NY 10004, USA. [12]Present address: Department of Surgery, McGill University, Montreal H4A 3J1 QC, Canada. [13]Present address: Cancer Research Program, The Research Institute of the McGill University Health Centre, Montreal H4A 3J1 QC, Canada. [14]Present address: Altos Labs, Redwood City, CA 94065, USA. ✉e-mail: Peng.Wu@cchmc.org; rnusse@stanford.edu

cancers with embryonic origins and relatively lower mutational burdens than adult cancers[6,7]. For instance, heterogeneity in neuroblastoma has been attributed to at least two differentiation states governed by distinct transcription factor networks[8,9], and a transition from the noradrenergic to the mesenchymal cell state leads to resistance to targeted immunotherapy[10].

Hepatoblastoma is the most common pediatric liver malignancy, posited to originate from hepatoblasts, the embryonic progenitor of the two main epithelial cell types in the liver: hepatocytes and cholangiocytes. Several features render this tumor type an excellent model to study the relative contributions of genetic and nongenetic drivers of heterogeneity. Genomic studies have revealed a remarkably low tumor mutational burden, with up to 80–90% of hepatoblastomas containing mutations of the *CTNNB1* gene that abrogate degradation of the β-catenin protein, leading to constitutive activation of the Wnt pathway[11–15]. Despite the relative simplicity of their genomic alterations, hepatoblastomas nevertheless demonstrate significant phenotypic and histologic heterogeneity. Individual hepatoblastomas are often comprised of a mixture of epithelial histologies, known as fetal and embryonal, as well as areas of mesenchymal differentiation[16]. Tumors comprised purely of well-differentiated fetal histology exhibit low mitotic activity and are cured with surgical resection alone[17], while those containing embryonal histology behave more aggressively. Thus, identifying biologic features that differentiate between the fetal and embryonal histologies has important treatment implications.

Previous studies using immunohistochemistry or microarrays on bulk tumor tissue have ascribed differential Wnt-driven transcriptional programs to tumors comprised primarily of fetal histology compared to those containing components of embryonal histology[18], so called C1 vs. C2 tumors, respectively[19]. Further studies have proposed additional molecular classification schemes known as the C1/C2A/C2B[20], MRS-1/MRS-2/MRS-3[21], hepatocytic/proliferative/mesenchymal[15], and hepatocytic/liver progenitor/mesenchymal[22] subgroups, which stratify tumors into different prognostic categories. However, due to the rarity of this cancer and technical difficulties in propagating tumor cells in vitro, research into the functional relevance of specific genes and biomarkers has been hindered by the limited number of patient-derived cell lines[23].

Although aberrant Wnt activation appears crucial to the development of hepatoblastoma, it has remained unclear how heterogeneity in the expression of Wnt target genes arises and what accounts for the differences in proliferation between the fetal and embryonal histologies. Prior studies of hepatoblastoma cell lines have attributed roles for Wnt/β-catenin[19], as well as the FGFs, FGF19, and FGF8, to cell proliferation[24,25]. Studies of normal hepatocytes suggest that Wnt and FGFs act at different steps in the cell cycle, as in vitro propagation of hepatocytes requires both exogenous Wnt agonists and growth factors acting through RTK/MAPK pathways[26,27]. In the normal liver, Wnt signals from endothelial cells in the central vein act on neighboring hepatocytes to promote mitosis[28]. Meanwhile, additional signals that drive hepatocyte cell cycle progression from G1 to S phase include the endocrine growth factor, FGF19[29], normally expressed in the small intestine and gall bladder in response to bile acids[30]. Thus, both Wnt and FGF signals are necessary for normal hepatocyte proliferation in vitro[26,27] and in vivo[29]. In the embryonic liver, Wnt/β-catenin promotes hepatoblast maturation and survival[31], while aberrant Wnt activation represses the hepatic fate and promotes biliary differentiation of hepatoblasts[32]. However, the implications of aberrant hepatobiliary differentiation for tumorigenesis in hepatoblastoma have not been reported.

Here, we use histology-guided RNA sequencing to elucidate the transcriptional heterogeneity within the fetal, embryonal, and mesenchymal components of hepatoblastoma. We uncover differentiation along the biliary and hepatic lineages as a key determinant of intra-tumoral heterogeneity. Through functional studies in prospectively obtained patient-derived tumoroids, we identify FGF19 as a context-specific target of Wnt in tumor cells with biliary differentiation marked by the embryonic transcription factor SOX4. FGF19, downstream of Wnt/β-catenin and SOX4, provides a requisite proliferative signal for surrounding tumor cells and promotes cell cycle progression together with constitutive Wnt activation.

## Results

### Histology-based RNA sequencing identifies increased expression of Wnt target genes, and proliferative and biliary markers in embryonal hepatoblastoma

To characterize the transcriptional profiles of distinct histologic components of hepatoblastoma, we used laser capture microdissection to isolate regions identified by a pathologist as embryonal, fetal, and mesenchymal histology, as well as adjacent normal liver, from a series of 17 patients with hepatoblastoma. The median age of these patients was 24 months (range: 5–83 months) and 47% were male (Supplementary Table S1). Of the tumor specimens, 53% showed epithelial (fetal and embryonal) histology, and 47% displayed mixed (mesenchymal and epithelial) histology (Supplementary Table S1). In total, we performed RNA sequencing on 74 microdissected samples using Smart-3SEQ[33], which permits accurate quantification of transcript abundance from small, formalin-fixed, paraffin-embedded (FFPE) specimens (Fig. 1a, Supplementary Fig. S1a, Supplementary Data 1). In the initial quality control analysis, two samples were outliers, with low RNA yield and less than 100,000 uniquely mapped reads; these were excluded from the final analyses (Supplementary Fig. S1b). For the remaining 72 samples, the percentage of uniquely mapped reads was comparable to the range of 17-27% previously reported for Smart-3SEQ of FFPE specimens[33] (Supplementary Fig. S1b).

Using unsupervised hierarchical clustering and principal component analysis to represent the transcriptional profiles of each sample in two dimensions, we found that the samples clustered based on their histologies rather than by patient or other patient-specific characteristics such as age, sex, or treatment history (Fig. 1b, Supplementary Fig. S1c). The transcriptional profiles of normal liver from either the pericentral or midlobular zones clustered together (Fig. 1b), thus these samples were all included as normal liver in further analyses. Similarly, mesenchymal components, dissected from areas that appeared to be osteoid, stromal, or cartilaginous, clustered together (Fig. 1b) and were collectively assigned as mesenchymal histology for additional analyses. Among the tumor components, the transcriptional profiles of specimens with fetal histology most resembled those of normal liver, while those with mesenchymal histology diverged the most from normal liver (Fig. 1b).

We next identified differentially expressed genes comparing the embryonal, fetal, and mesenchymal tumor components to normal liver, using DESeq2[34] (Fig. 1c, Supplementary Table S2, Supplementary Data 2). Using a threshold of $\log_2$ fold change > 1 and < −1 and padj <0.05 for statistical significance, we found that all the tumor components shared a common set of 265 upregulated and 337 downregulated genes (Fig. 1c). The embryonal and mesenchymal components shared the most differentially expressed genes in common (508 upregulated, 470 downregulated), while the fetal and mesenchymal histologies shared the fewest differentially expressed genes (24 upregulated, 17 downregulated) (Fig. 1c).

Gene set enrichment analysis (GSEA)[35,36] revealed significant upregulation of Wnt/β-catenin signaling in all tumor components as compared to normal liver (Supplementary Fig. S1d). Both the fetal and embryonal components showed upregulation of cell cycle regulators such as E2F and MYC targets as well as G2M checkpoint components (Supplementary Fig. S1d). Unique to the embryonal component were DNA repair and mitotic spindle genes (Supplementary Fig. S1d). Meanwhile, the mesenchymal components showed upregulation of

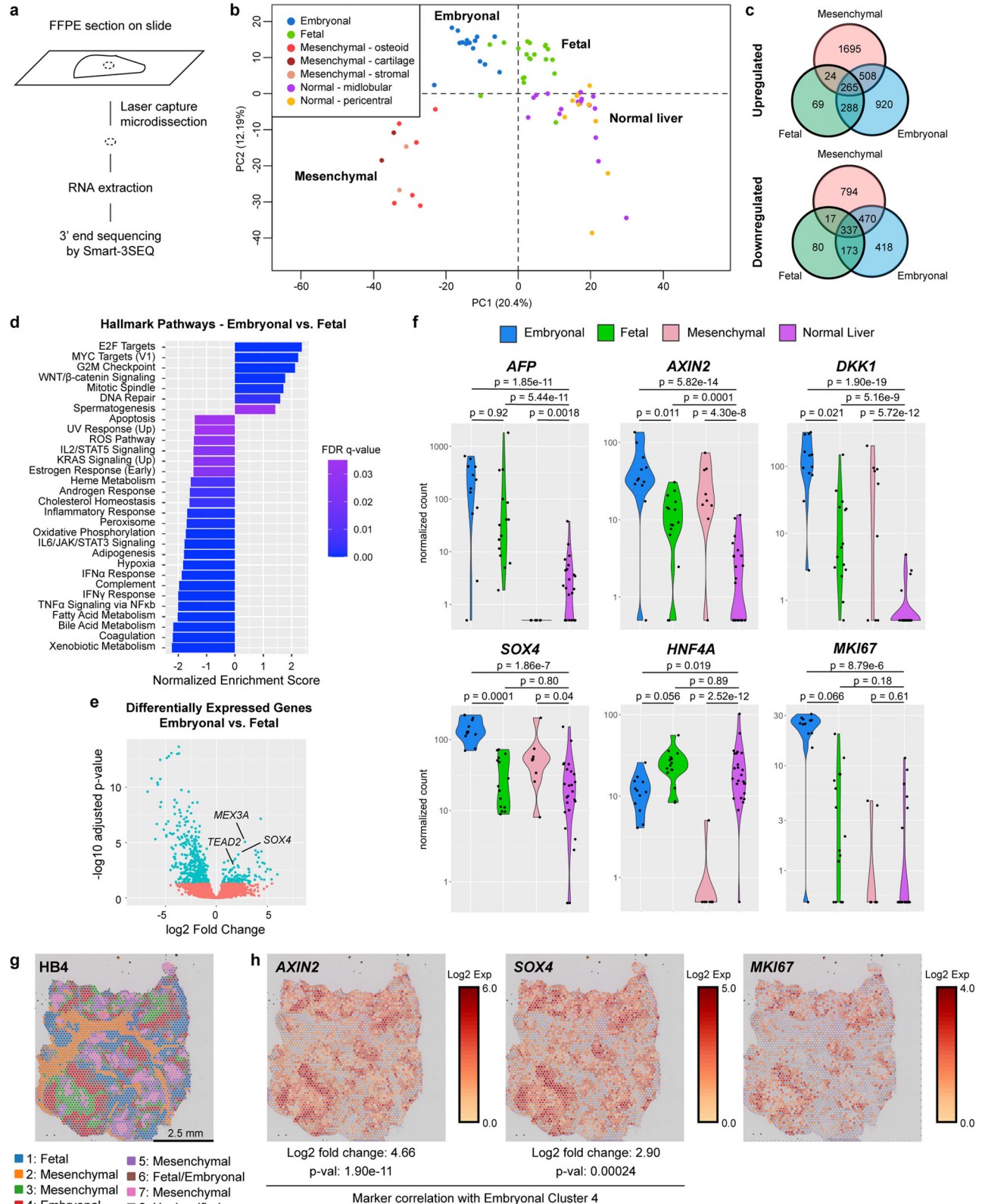

epithelial-mesenchymal transition and TGFβ signaling (Supplementary Fig. S1d). Comparing the embryonal to the fetal components of hepatoblastoma (Supplementary Data 2), upregulated pathways included Wnt signaling, MYC targets, E2F targets, mitotic spindle, G2M checkpoint, and DNA repair genes (Fig. 1d). The list of genes significantly upregulated in the embryonal compared to fetal components (Supplementary Table S2, Supplementary Data 2) also included

SOX4, a transcription factor important for biliary differentiation[37,38], and two of its reported target genes, TEAD2[39] and MEX3A[40] (Fig. 1e-f).

Focusing further on specific genes, we found that Wnt targets, such as AXIN2 and DKK1, were significantly upregulated in all components of the tumors, but particularly increased in the embryonal components (Fig. 1f). In addition to higher expression of Wnt target genes, embryonal components showed increased expression of

**Fig. 1 | Histology-based RNA sequencing reveals increased expression of Wnt target genes, proliferation, and cholangiocyte markers in embryonal hepatoblastoma. a** Schematic of Smart-3SEQ. **b** Principal component analysis plot of RNA sequencing from 72 samples with the histologies noted, using top 1500 expressed genes. **c** Venn diagram of differentially expressed, upregulated and downregulated, genes (log₂ fold change > 1 or < −1, padj <0.05 by two-tailed Wald test with Benjamini-Hochberg adjustment for multiple hypothesis testing) in embryonal, fetal, or mesenchymal hepatoblastoma compared to normal liver, determined by DESeq2. **d** Top hallmark pathways differentially expressed between embryonal and fetal components of hepatoblastoma (FDR < 0.05), using gene set enrichment analysis (GSEA) on pre-ranked gene lists obtained by DESeq2. **e** Volcano plot of differentially expressed genes between embryonal and fetal components of hepatoblastoma with SOX4 and target genes labeled. Note: range of axes chosen for

clarity, with one outlier differentially upregulated gene falling beyond the plotted axes. Adjusted p-values determined by two-tailed Wald test with Benjamini-Hochberg adjustment for multiple hypothesis testing. **f** Normalized counts of representative markers of Wnt pathway, proliferation, and cholangiocytic differentiation detected by Smart-3SEQ. Adjusted p-values determined by two-tailed Wald test with Benjamini-Hochberg adjustment for multiple hypothesis testing. Blue: embryonal, green: fetal, pink: mesenchymal, purple: normal liver. **g** Clusters identified by 10x Visium spatial transcriptomics on primary hepatoblastoma HB4, assigned to fetal, embryonal, or mesenchymal histologies based on genes identified by Smart-3SEQ. **h** Spatial expression of *AXIN2*, *SOX4*, and *MKI67* in HB4. Adjusted p-values determined in CellRanger by negative binomial test with Benjamini-Hochberg adjustment for multiple hypothesis testing. Source data are provided as a Source Data file. See also Supplementary Table S1, S2, Supplementary Fig. S1.

markers of cell cycle progression, such as *MKI67* and *CCND2* (Fig. 1f, Supplementary Fig. S1e). As compared to normal liver, the embryonal components also showed higher expression of the biliary marker *KRT19* (Supplementary Fig. S1e), in addition to the previously mentioned biliary transcription factor *SOX4* (Fig. 1f), and relatively lower expression of the hepatic transcription factor *HNF4A* (Fig. 1f). Meanwhile, other known markers of hepatoblastoma, including *AFP*, *DLK1*, and *GPC3*, were upregulated in the epithelial (similarly in both fetal and embryonal) but not mesenchymal components (Fig. 1f, Supplementary Fig. S1e).

We further validated these findings using spatial transcriptomic analyses of tumor tissue from two patients not included in the Smart-3SEQ cohort (HB4 and HB17, patient details in Supplementary Table S3). By identifying the overlap between marker genes specific for each cluster and those identified by Smart-3SEQ as specific for the fetal, embryonal, and mesenchymal components of hepatoblastoma, nearly all the clusters could be assigned as either fetal, embryonal, or mesenchymal (Fig. 1g, Supplementary Fig. S2a). Specifically, *AXIN2* and *SOX4* were among the differentially expressed genes identified in the clusters that correlated with embryonal histology (cluster 4 in HB4 and cluster 1 in HB17; Fig. 1h, Supplementary Fig. S2b, cluster markers listed in Supplementary Data 3), while showing only scattered low expression in normal liver (Supplementary Fig. S1f). The embryonal regions further showed high expression of *MKI67* (Fig. 1h, Supplementary Fig. S2b). Thus, both the Smart-3SEQ and spatial transcriptomic data confirmed that the embryonal components of hepatoblastoma corresponded to higher expression of Wnt target genes, biliary markers, and proliferation markers. We subsequently focused on understanding the relationship between these features in embryonal hepatoblastoma.

## FGF19 is focally expressed within highly proliferative regions of embryonal hepatoblastoma

Next, we examined the expression of candidate drivers of increased proliferation in embryonal hepatoblastoma. Interestingly, several FGFs, including *FGF8*, *FGF9*, and *FGF19*, were expressed at low levels in the embryonal components but not in the fetal components or in the normal liver (Fig. 2a-c, Supplementary Fig. S1f, S2c-f). In contrast, the growth factor *IGF2*, known to be upregulated in hepatoblastomas[41], was similarly elevated in both the fetal and embryonal components (Supplementary Fig. S2f). Other growth factors expressed in normal liver, such as *IGF1* and *HGF*, were not elevated in hepatoblastoma compared to normal liver, while *EGF* was not detected in the tumor regions or normal liver (Supplementary Fig. S2f).

We were particularly interested in *FGF19* as its secretion by a hepatoblastoma cell line has been previously reported to serve as an autocrine growth factor[24]. Furthermore, in mice, pericentral Wnt-responsive hepatocytes proliferate in response to FGF15, the mouse ortholog of FGF19, which is secreted by the small intestine[42] during cycles of fasting and re-feeding and leads to cell proliferation in conjunction with Wnt signals[29]. Among 12 independent embryonal tumor

components (from different patients) that were dissected and analyzed by Smart-3SEQ, 9 specimens (75%) showed evidence of FGF19 expression by RNA sequencing (Fig. 2a, Supplementary Table S1). By contrast, FGF19 expression was detected by Smart-3SEQ in only 1 of 15 independent fetal tumor specimens (7%) and 1 of 15 normal liver specimens (7%) (Fig. 2a, Supplementary Table S1). Similarly, by spatial transcriptomics, *FGF19⁺* spots were overrepresented in the embryonal clusters in both HB4 and HB17, accounting for ~7-8% of the embryonal cluster while representing only ~2% of the tumor as a whole (Fig. 2b-c, Supplementary Fig. S2c-e). As a control, spatial transcriptomic analyses of a section of normal liver from a patient with hepatoblastoma showed very low expression of *FGF19*, *AXIN2*, and *SOX4*, without apparent co-localization (Supplementary Fig. S1f).

We confirmed FGF19 expression by RNA in situ hybridization in 6 of 7 available specimens with upregulation of FGF19 in the embryonal components by Smart-3SEQ (Fig. 2d, Supplementary Fig. S3a-c, S4a, summarized in Supplementary Table S1). Strikingly, cells expressing *FGF19* characteristically clustered within regions of embryonal histology with multiple *FGF19*-expressing foci appearing in the same tumor, at varying distances from each other (Fig. 2d, Supplementary Fig. S3a-c, S4a). The *FGF19*-expressing foci localized to a subset of cells within regions showing high expression of *AXIN2* and *SOX4* (Fig. 2e-f, Supplementary Fig. S2g). On further analyses of double in situ hybridization for both *FGF19* and either *AXIN2* or *SOX4*, FGF19-expressing cells comprised approximately 4% and 10% of *AXIN2⁺* and *SOX4⁺* cells, respectively, consistent with the fact that nearly all tumor cells were *AXIN2⁺* while only a fraction of tumor cells were *SOX4⁺* (Fig. 2g). There were almost no *FGF19*-expressing cells that did not also express either *AXIN2* or *SOX4* (Fig. 2h), suggesting that *FGF19* expression is a unique feature of a subset of *AXIN2⁺*, *SOX4⁺* tumor cells.

To determine which cells might be responding to the FGF19 signal, we investigated the expression pattern of *KLB*, the co-receptor required for FGF19 signaling through FGFR4, whose expression is more restricted and representative of active signaling than expression of FGFR4[43]. Interestingly, *KLB* did not co-localize with *FGF19* but was highly expressed in cells surrounding the *FGF19*-expressing foci (Fig. 2f-h), suggesting that FGF19 might serve as a paracrine signal.

We next interrogated the correlation of *FGF19* expression in primary hepatoblastoma tissues with the proliferative index and pattern of β-catenin staining, where nuclear β-catenin indicates high levels of Wnt signaling, while normal hepatocytes exhibit primarily membrane-bound β-catenin. Immunofluorescence for β-catenin in hepatoblastomas showed three patterns: cells with low nuclear and primarily membranous staining (Low in Fig. 3a), those with intermediate nuclear staining (Med in Fig. 3a), and foci of cells with high nuclear staining (High in Fig. 3a) that co-localized with regions of *FGF19* expression in serial sections (Fig. 3b). Interestingly, cells with intermediate nuclear β-catenin staining (Med) surrounded the *FGF19*-expressing cells and were more likely to be positive for MKI67 (marker of active cell cycling)

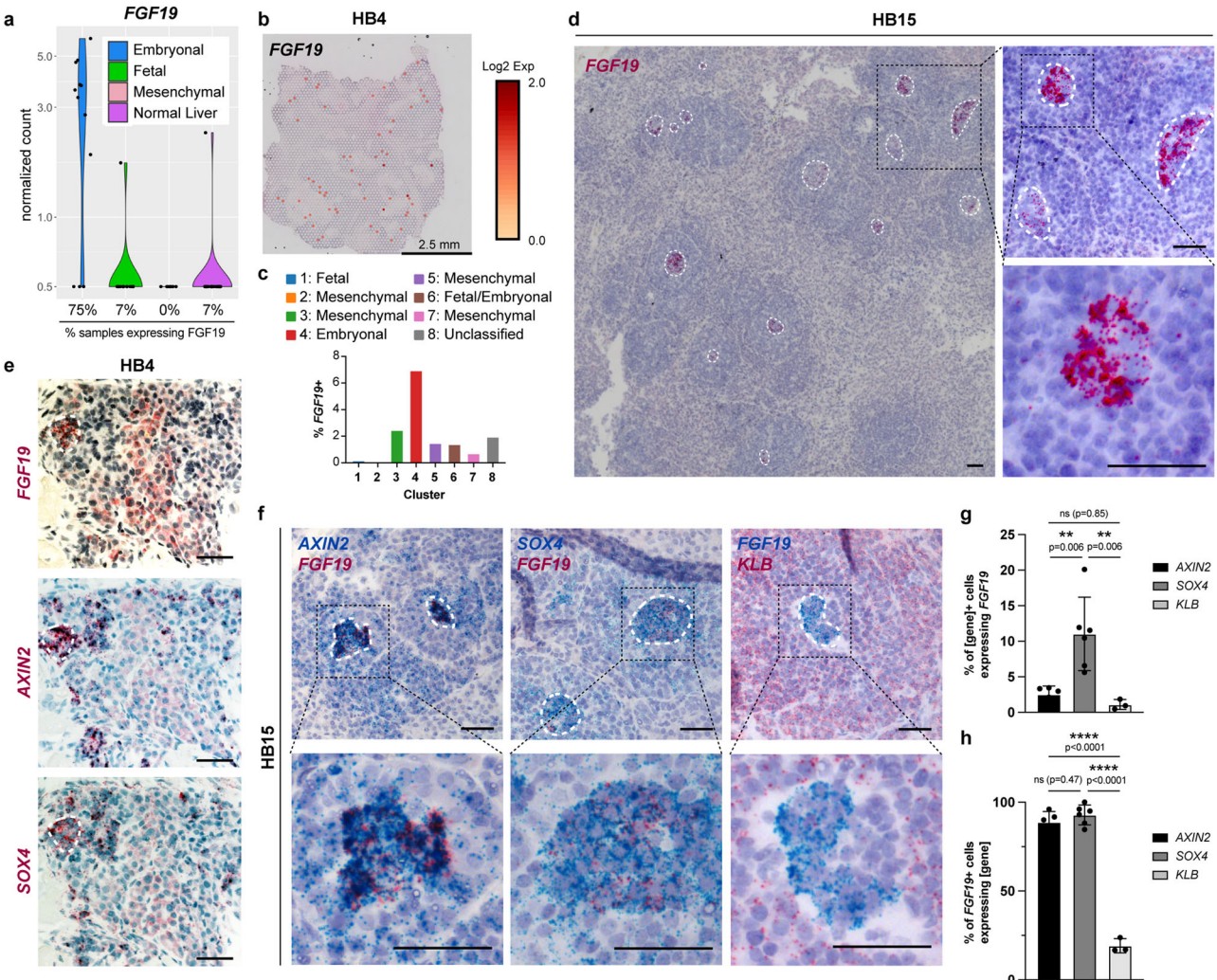

**Fig. 2 | FGF19 is expressed focally within proliferative areas of embryonal hepatoblastoma. a** Normalized counts of *FGF19* detected by Smart-3SEQ. **b** Spatial expression of *FGF19* in HB4 by 10x Visium. Adjusted p-values determined in Cell-Ranger by negative binomial test with Benjamini-Hochberg adjustment for multiple hypothesis testing. **c** Quantification of *FGF19*⁺ spots/total spots in each cluster. **d** RNA in situ hybridization in primary hepatoblastoma HB15 detecting *FGF19* (red) at low and high magnification. White dashed lines outline cells expressing *FGF19*. **e** Representative RNA in situ hybridization in serial sections of primary hepatoblastoma HB4 detecting *FGF19*, *AXIN2*, and *SOX4* in red. White dashed lines outline cells expressing *FGF19*. Similar results obtained for *n* = 4 different patient specimens. **f** Double RNA in situ hybridization of *AXIN2* (blue) / *FGF19* (red), *SOX4* (blue) /

*FGF19* (red) and *FGF19* (blue) / *KLB* (red) in HB15 at low and high magnification. Similar results obtained for *n* = 4 different patient specimens. **g** Quantification of cells expressing *AXIN2* (*n* = 5 images, 3074 cells), *SOX4* (*n* = 6 images, 1441 cells), or *KLB* (*n* = 3 images, 1935 cells) that also co-express *FGF19* as detected in **f**. Mean and s.d, ordinary one-way ANOVA with Tukey's correction for multiple comparisons. **h** Quantification of cells expressing *FGF19* that also co-express *AXIN2* (*n* = 5 images, 84 cells), *SOX4* (*n* = 6 images, 161 cells), or *KLB* (*n* = 3 images, 101 cells) as detected in **f**. Mean and s.d., ordinary one-way ANOVA with Tukey's correction for multiple comparisons. For all panels – scale bar: 50 µm unless otherwise indicated, *: *p* < 0.05, **: *p* < 0.01, ***: *p* < 0.001, ns: not significant. Source data are provided as a Source Data file. See also Supplementary Fig. S2, S3, S4.

and phospho-histone H3 (marker of mitosis) than the cells with either high or low nuclear β-catenin staining (Fig. 3a-c, Supplementary Fig. S5a-g).

Given the correlation between the expression of *FGF19* and the biliary transcription factor *SOX4*, we investigated the differentiation state of these cells by co-immunofluorescence for the hepatic marker HNF4A and the biliary marker KRT19 (Fig. 3d). We found that foci of *FGF19* expression co-localized with tumor cells that were HNF4A⁻KRT19⁺, which we denote as cholangiocytic (Fig. 3d). Upon quantification, such cholangiocytic cells were more frequently found in embryonal as compared to fetal regions of hepatoblastoma (Fig. 3e, Supplementary Fig. S5h). Meanwhile, in the embryonal regions, there were significantly lower percentages of HNF4A⁺KRT19⁻ (hepatocytic) cells, while both fetal and embryonal components contained similar percentages of HNF4A⁺KRT19⁺ (hepatoblastic) cells and low frequency

of HNF4A⁻KRT19⁻ (non-hepatobiliary) cells (Fig. 3d-e, Supplementary Fig. S5h). Co-localization of KRT19 and MKI67 showed a high proliferative index among hepatoblastic cells compared to low (but not absent) proliferation of the cholangiocytic and hepatocytic cells (Supplementary Fig. S5i). Thus, *FGF19* expression correlates with a cholangiocytic differentiation state.

## Patient-derived tumoroids provide a system to test growth requirements of hepatoblastoma

To develop an experimental system to test the functional requirements of the different signaling pathways in hepatoblastoma, we obtained fresh tumor specimens from patients and generated primary 3D tumoroids[44]. Cell clusters were embedded in Matrigel and propagated in media containing the growth factors, EGF, FGF10, and HGF. Of note, the media did not include exogenous Wnt agonists, forskolin, or

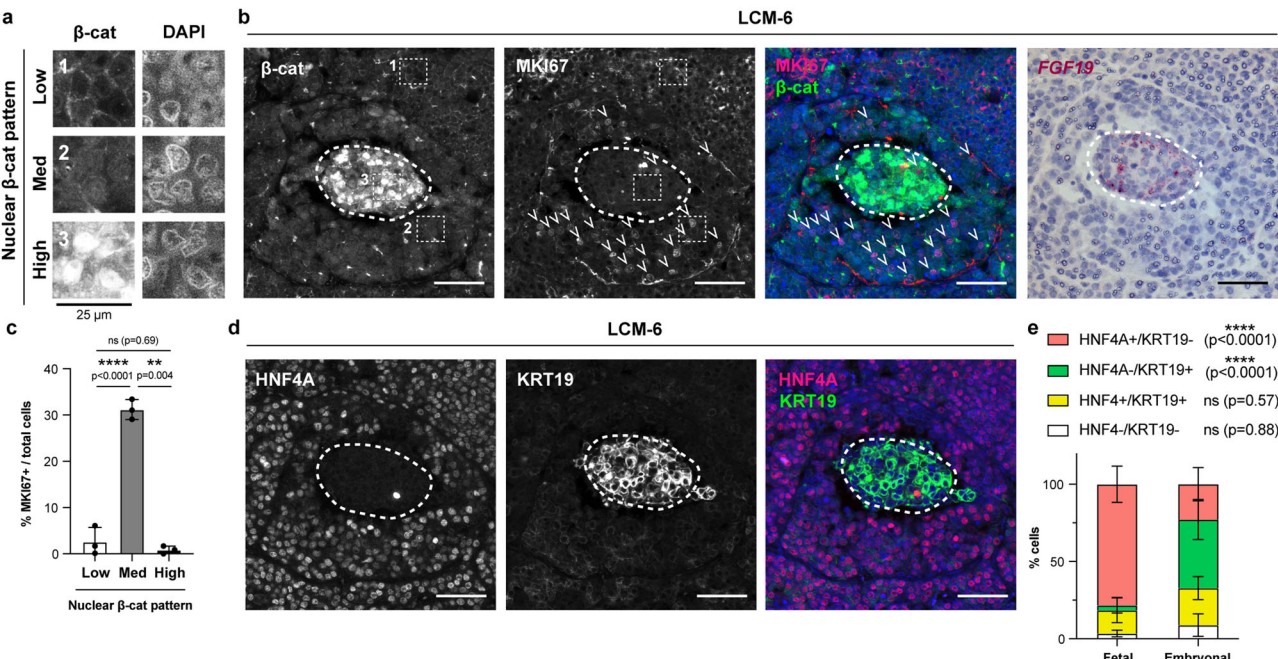

**Fig. 3 | *FGF19*-expressing foci are distinguished from surrounding cells with high proliferative activity by different patterns of nuclear β-catenin and expression of hepatobiliary markers. a** High magnification of three patterns of nuclear β-catenin localization (1: Low, 2: Med, 3: High) corresponding to numbered regions outlined with white dotted squares in **b**. scale bar: 25 μm. **b** Serial sections of primary hepatoblastoma LCM-6 showing (from left): Co-immunofluorescence of β-catenin (green) and MKI67 (magenta), merged with DAPI (blue); RNA in situ hybridization of *FGF19*. White dashed lines outline cells expressing *FGF19*. White arrowheads indicate MKI67+ cells. **c** Quantification of the percentage of MKI67+ cells with nuclear β-catenin staining patterns as shown in **a**. Mean and s.d., $n = 3$ patient specimens, >1000 cells scored for each specimen, ordinary one-way ANOVA, paired, with Tukey's correction for multiple comparisons. **d** Co-

immunofluorescence for HNF4A (magenta) and KRT19 (green), merged with DAPI (blue) in serial sections of primary hepatoblastoma LCM-6 as in **b**. White dashed lines outline cells expressing FGF19. **e** Quantification of percentage of total cells with HNF4A$^+$KRT19$^-$ (hepatocytic, pink), HNF4A-KRT19$^+$ (cholangiocytic, green), HNF4A$^+$KRT19$^+$ (hepatoblastic, yellow), and HNF4A-KRT19$^-$ (non-hepatobiliary, white) staining patterns in fetal ($n = 4$ patient specimens) and embryonal ($n = 3$) components of hepatoblastoma. Mean and s.d., >100 cells scored for each specimen, fetal vs. embryonal compared by ANOVA mixed effects, with Sidak's correction for multiple comparisons. For all panels – scale bar: 50 μm unless otherwise indicated, **: $p < 0.01$, ****: $p < 0.0001$, ns: not significant. Source data are provided as a Source Data file. See also Supplementary Fig. S5.

TNF-α, which have previously been used in the culture of normal hepatocytes[26,27].

From a series of 16 patients, we successfully obtained 10 independent cultures that could be serially passaged as tumoroids for >100 days (Supplementary Table S3). Morphologically, the tumoroids formed spherical cell clusters or occasionally multilobulated structures (Fig. 4a-b). The patients from whom these tumoroids derived were representative of the general population of patients with hepatoblastoma, with a median age of 19 months (range: 4-129 months) and a slight male predominance (62%). Four of the tumoroids were obtained prior to any chemotherapy, while the remaining tumoroids were isolated at the time of resection following neoadjuvant chemotherapy (Supplementary Table S3).

We first investigated the status of the *CTNNB1* gene in the tumoroids, since activating Wnt pathway mutations have been described in the majority of hepatoblastomas[11–15], and all the tumoroids grew in media without an exogenous Wnt agonist. We found evidence of mutations involving exon 3 of *CTNNB1* in all of the tumoroids, ranging from deletion of the entire exon to point mutations in residues important for GSK3β phosphorylation and subsequent targeting of β-catenin for ubiquitin-mediated degradation (Fig. 4c, Supplementary Fig. S6a). Targeted cancer gene panel sequencing of the primary tumor specimens identified a few additional mutations, including one patient each with pathogenic germline mutations in *APC* and *ARID1A*, while the remaining mutations were variants of unknown significance (Supplementary Table S3).

Growth analysis of a subset of tumoroids revealed a doubling time between 41-126 hours (Fig. 4d). All tumoroids at early passage showed

expression of known hepatoblastoma markers, *AFP*, *DLK1*, and the Wnt target gene, *AXIN2* (Fig. 4e-g). However, the tumoroids differed in their relative expression of hepatocyte and cholangiocyte lineage markers, with some tumoroids (HB15 and HB17) showing lower expression of *HNF4A* and higher expression of *SOX4* as compared to others (HB1) (Fig. 4e). Interestingly, the relative expression of hepatobiliary lineage markers correlated with the expression of *FGF19* in the tumoroids. Tumoroids with high *HNF4A* and low *SOX4* (eg. HB1) showed no evidence of FGF19 production, while those with higher expression of *SOX4* (HB15 and HB17) exhibited robust FGF19 secretion (Fig. 4j).

Immunofluorescence for hepatocyte and cholangiocyte markers showed that tumoroids with relatively high *HNF4A* expression (eg. HB1) contained cells that co-expressed both hepatocytic and cholangiocytic markers and maintained expression of these markers over multiple passages (Fig. 4f, h, Supplementary Fig. S6b). Similarly, expression of the Wnt target gene, *TBX3*, was also maintained at early and late passages (Fig. 4f, h). On the other hand, those tumoroids with lower expression of *HNF4A* (eg. HB15) showed heterogeneous expression of hepatocytic and cholangiocytic markers (Fig. 4g, i). At later passages, these tumoroids lost expression of *HNF4A* and *AFP*, while almost exclusively expressing *KRT19* (Fig. 4g, i, Supplementary Fig. S6b). This suggested that the *FGF19*-expressing, cholangiocytic tumor cells might have a selective advantage in our culture system over passages. Given this apparent evolution in a fraction of the tumoroid cultures over serial passages, we limited our subsequent experiments to early passage (< P4) tumoroids.

Finally, to determine whether the pattern of *FGF19* and *KLB* expression in tumoroids resembled that of primary tumors, we

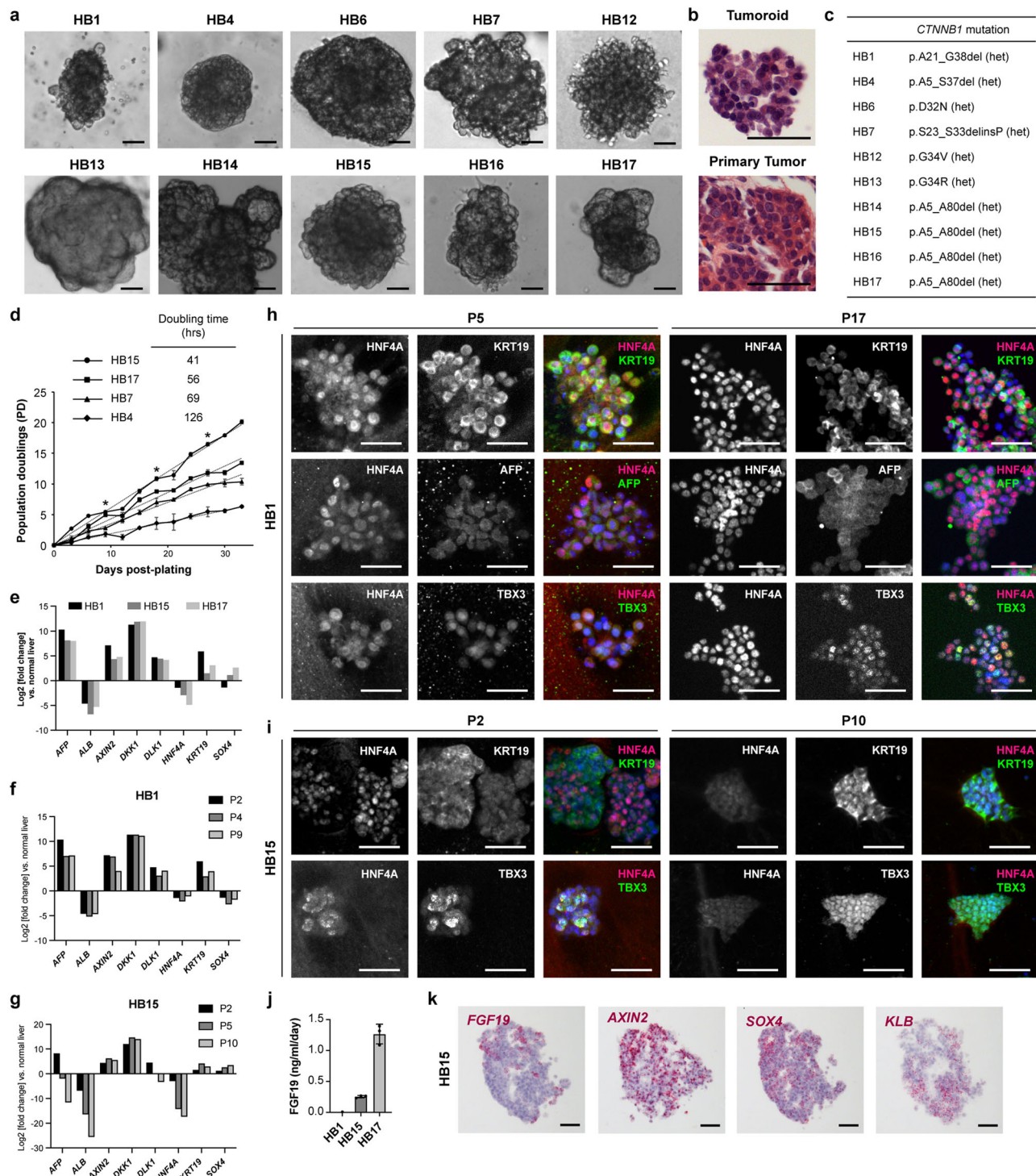

performed RNA in situ hybridization on the *FGF19*⁺ tumoroids, which revealed *FGF19* expression surrounded by cells expressing *KLB* (Fig. 4k). *AXIN2* and *SOX4* were highly expressed in these tumoroids (Fig. 4k), and *FGF19* expression comprised a fraction of the *AXIN2*⁺ or *SOX4*⁺ cells, similar to the case in primary tissues (Fig. 4k).

### Transcriptional heterogeneity of hepatoblastoma tumoroids corresponds to the embryonal and fetal signatures identified by Smart-3SEQ

Given the apparent transcriptional heterogeneity in the hepatoblastoma tumoroids, as detected by qRT-PCR and in situ hybridization, we sought to more precisely characterize gene expression in the tumoroids by single cell RNA sequencing of dissociated cells from early passage tumoroids. We analyzed gene expression data for a total of 40,666 cells from P1-P2 tumoroids corresponding to the 10 different patients and clustered the cells using Seurat and UMAP visualization[45] (Fig. 5a-b, Supplementary Data 4). Clusters 0, 2, and 4, characterized by high expression of known hepatoblastoma markers, including *AFP*, *DLK1*, and *GPC3*, were shared among all the tumoroids, while the remaining clusters of cells were more heterogeneously represented across the different tumoroids (Fig. 5a-c, Supplementary Fig. S7a-c, Supplementary Data 4).

Focusing on the differentially expressed genes in each cluster, we found that clusters 3 and 10 were marked by upregulation of *FGF19*, as

**Fig. 4 | Patient-derived hepatoblastoma tumoroids recapitulate features of primary tumors. a** Representative brightfield images of primary hepatoblastoma tumoroids at early passage (P1-2) from patients indicated. **b** Representative H&E image of early passage tumoroid compared to the primary tumor from the same patient. Similar results obtained for *n* = 5 different patient specimens. **c** *CTNNB1* mutations in the hepatoblastoma tumoroids. **d** Growth curve of several tumoroids starting from early passage (P1-P2). Asterisks denote timepoints of passaging of all tumoroids except for HB7, which was passaged every 6 days instead of every 9 days. Mean and s.d., *n* = 3 experiments for each time point. **e** Expression of hepatoblastoma and hepatobiliary marker genes in early passage (P1-P2) tumoroids by qRT-PCR, compared to normal liver. **f** Expression of hepatoblastoma and hepatobiliary marker genes by qRT-PCR, compared to normal liver, in FGF19-negative tumoroid, HB1, at different passages, P2, P4, and P9, as indicated. **g** Expression of hepatoblastoma and hepatobiliary marker genes by qRT-PCR, compared to normal liver, in FGF19-positive tumoroid, HB15, at different passages, P2, P5, and P10, as

indicated. **h** Immunofluorescence detection of hepatobiliary markers (*HNF4A* in magenta, *KRT19* in green, *AFP* in green) and Wnt target gene *TBX3* (green) in a representative FGF19-negative tumoroid, HB1, at early (P5) and late (P17) passage. Similar results obtained on *n* = 4 different FGF19-negative patient tumoroids. **i** Immunofluorescence detection of hepatobiliary markers (*HNF4A* in magenta, *KRT19* in green, *AFP* in green) and Wnt target gene *TBX3* (green) in a representative FGF19-positive tumoroid, HB15, at early (P2) and late (P10) passage. Similar results obtained on *n* = 3 different FGF19-positive patient tumoroids. **j** ELISA detecting FGF19 in the tumoroids shown in **e**, reported as ng/ml of media/day per 100,000 cells plated. Mean and s.d., *n* = 3 independent experiments. **k** RNA in situ hybridization detecting *FGF19*, *KLB*, *SOX4*, and *AXIN2* in red in serial sections of a representative HB15 tumoroid. Similar results obtained on *n* = 3 different FGF19-positive patient tumoroids. For all panels – scale bar: 50 µm. Source data are provided as a Source Data file. See also Supplementary Table S3, Supplementary Fig. S6.

well as *AXIN2* and *SOX4* (Fig. 5c, Supplementary Fig. S7a, c, Supplementary Data 4), analogous to the *FGF19*-expressing cells that we identified in the primary hepatoblastomas (Fig. 2a-h). These cells also showed high expression of *FGF3*, *FGF8*, and *FGF9* (Supplementary Fig. S7b-c). Cycling tumor cells, marked by *MKI67*, primarily belonged to clusters 2, 4, and 10 (Fig. 5c, Supplementary Fig. S7a, c). A subset of the *MKI67*-expressing cells in clusters 2 and 4 overlapped with the expression of the receptor and co-receptor, *FGFR4* and *KLB* (Fig. 5c, Supplementary Fig. S7a-c). This pattern of high *FGF19* expression in a population of cells distinct from those expressing the cognate receptors resembled the expression pattern in primary tumors (Fig. 2f-h), again suggesting that FGF19 promotes proliferation in a paracrine fashion, acting on cells other than the FGF19-producing cells themselves. Clusters defined by the expression of the marker genes identified by integrated analyses were similarly found through separate analyses of the data from each tumoroid (Supplementary Fig. S8a-j). These analyses further confirmed that the tumoroids segregated between those containing FGF19-expressing cells (HB4, HB7, HB15, HB17) and those that had almost no evidence of FGF19 expression (HB1, HB6, HB12, HB13, HB14, HB16) (Supplementary Fig. S7d, S8a-j).

We next determined whether the transcriptional profiles of the hepatoblastoma cells in vitro recapitulated the histology-based transcriptional signatures identified in the primary tumors using Smart-3SEQ. With the top 50 differentially upregulated genes in the fetal or embryonal components of hepatoblastoma compared to normal liver, as determined by Smart-3SEQ (gene lists in Supplementary Table S2, Supplementary Data 5), we calculated a fetal or embryonal signature, respectively, for each cell, using the AddModuleScore function in Seurat (Fig. 5d). As there was significant overlap in cells expressing both fetal and embryonal signatures (in clusters 0, 2, 4, and 8), we next used the top 50 differentially upregulated or downregulated genes in the fetal compared to embryonal components (gene lists in Supplementary Table S2, Supplementary Data 5) to identify clusters expressing a fetal-specific or embryonal-specific gene signature, respectively (Fig. 5d). While the fetal-specific signature was highest in a subset of cell clusters 0 and 1, the embryonal-specific signature was highest in clusters 3, 6, 8, 9, and 10 (Fig. 5d).

Next, we calculated a mesenchymal signature for each cell based on the top 50 genes differentially upregulated in the mesenchymal components compared to the normal liver (Fig. 5d, gene list in Supplementary Table S2, Supplementary Data 5). Although there was significant overlap among cells expressing the mesenchymal signature with those expressing the embryonal-specific signature, almost no cells showed high expression of a mesenchymal-specific signature, derived from genes differentially upregulated in the mesenchymal compared to the embryonal components of the hepatoblastomas (Fig. 5d, gene list in Supplementary Table S2, Supplementary Data 5). Furthermore, while we detected *VIM* and *MSX2*, previously reported mesenchymal hepatoblastoma markers[22], as highly upregulated in the

same cells that had a high embryonal-specific signature, we failed to detect other classical mesenchymal markers such as *TWIST1*, *SNAI1*, or *SNAI2*. Overall, this data suggested that the cultured cells primarily represented the epithelial components of primary hepatoblastomas, encompassing cells with both fetal and embryonal patterns of gene expression, the latter of which contains some mesenchymal features. However, these tumoroids did not appear to contain pure mesenchymal cells.

We next performed RNA velocity[46] and pseudotime analyses[47] to identify possible differentiation trajectories of the tumoroid cells (Fig. 5e). RNA velocity on a subset of tumoroids showed that cells expressing markers common to both fetal and embryonal populations appeared to differentiate towards either the fetal-specific or embryonal-specific fates (Fig. 5e). Pseudotime analysis also corroborated that the fetal-specific and embryonal-specific cell fates may correspond to differentiation from progenitor cells expressing markers common to both fetal and embryonal tumor cells (Fig. 5e).

Compared to published studies, we found that cells expressing either the fetal or embryonal signatures overlapped with cells highly expressing upregulated genes previously reported in hepatoblastoma[19] (Fig. 5f, gene list in Supplementary Data 5). Meanwhile, cells with a higher fetal-specific signature correlated with the hepatoblast II gene signature previously obtained from single-cell RNA sequencing of primary tumor tissue from a patient with pure fetal hepatoblastoma[48], the Tr3 (maintenance) signature from single nuclei RNA sequencing of primary hepatoblastomas[49], and the hepatocytic subtype identified in a recent multi-omic study of hepatoblastoma[50] (Fig. 5f, gene lists in Supplementary Data 5). These fetal-specific cells showed weak expression of two hepatocytic hepatoblastoma gene signatures previously identified from bulk RNA sequencing[15,22] (Fig. 5f, gene list in Supplementary Data 5). Meanwhile, cells with the embryonal signature comprised a subset of cells with high expression of the C2 gene signature, previously attributed by microarray studies to embryonal hepatoblastoma and correlating with a worse outcome[19] (Fig. 5f, gene list in Supplementary Data 5). Finally, cells in clusters 3, 6, 9, 10 with high expression of both embryonal-specific and mesenchymal signatures correlated best with the neuroendocrine[48] and Tr5 (progression)[49] gene signatures identified previously by single cell and single nuclei RNA sequencing of primary hepatoblastomas and showed overlap with the liver progenitor class identified in recent multi-omic analyses[50] (Fig. 5f, gene lists in Supplementary Data 5). These embryonal-specific cells also correlated weakly with the mesenchymal hepatoblastoma gene signatures reported by Hirsch et al.[22] but with the proliferative rather than mesenchymal signature identified by Nagae et al.[15] (Fig. 5f, gene lists in Supplementary Data 5).

To better understand how the hepatoblastoma cells correlate to different stages of normal hepatoblast differentiation, we further analyzed the expression of markers of normal hepatoblasts, hepatocytes, and cholangiocytes, as defined by previously published

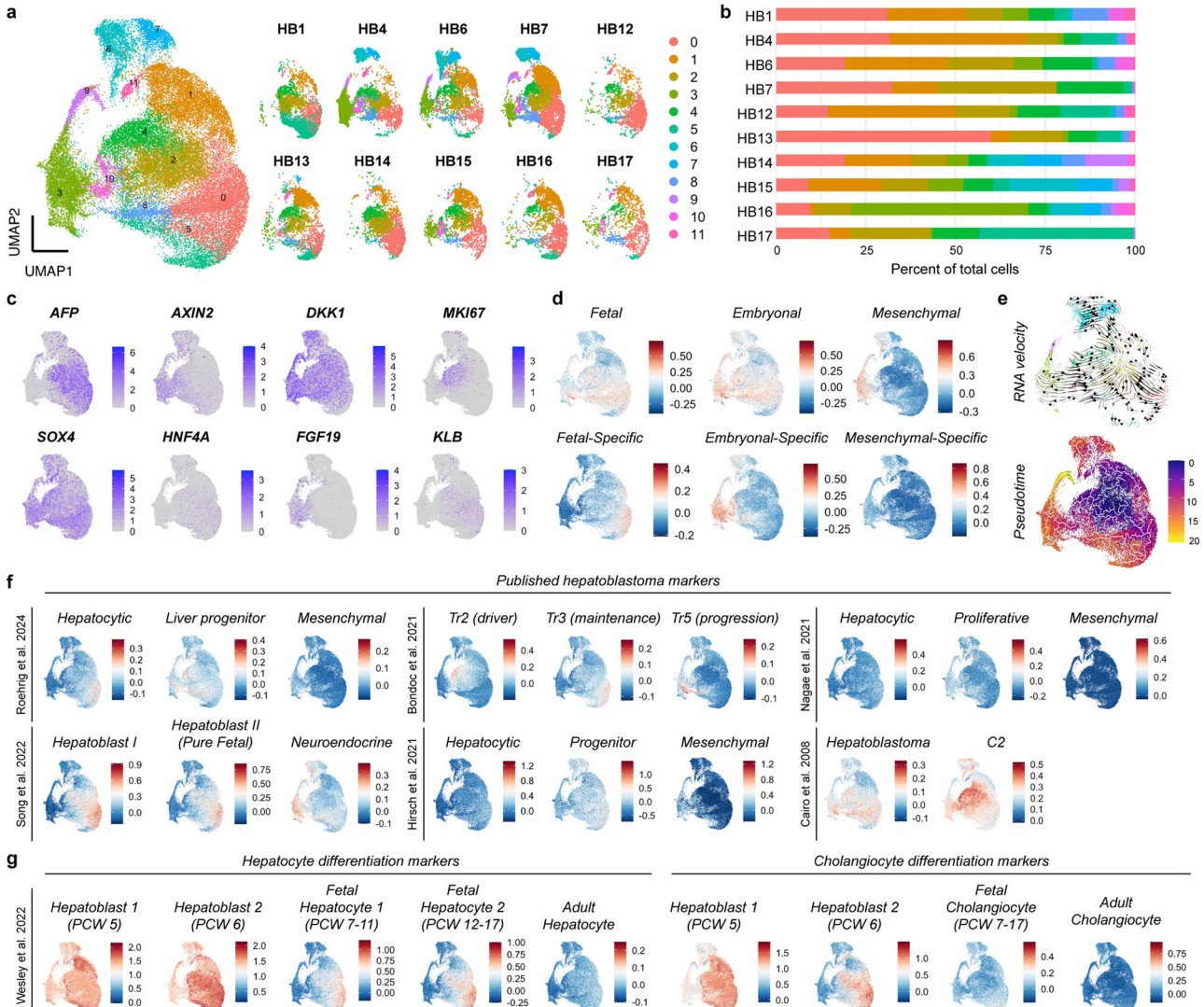

**Fig. 5 | Transcriptomic heterogeneity of hepatoblastoma tumoroids correlates with the embryonal and fetal gene signatures identified by Smart-3SEQ.**
**a** UMAP representation of scRNA sequencing results of 10 patients, colored by cluster, along with separate UMAP plots for each patient. **b** Bar graph showing representation of different UMAP clusters across 10 tumoroids. **c** UMAP plots showing relative gene expression for hepatoblastoma markers, Wnt target genes, proliferation and differentiation markers, and *FGF19* and its receptor/co-receptor, *FGFR4/KLB*. Color scales indicate normalized expression obtained by the Log-Normalize function in Seurat. **d** UMAP plots showing relative expression of different gene signatures defined as follows and listed in Supplementary Table S2, and Supplementary Data 5: fetal = top 50 differentially upregulated genes between the fetal histology vs. normal liver; embryonal = top 50 differentially upregulated genes between the embryonal histology vs. normal liver; fetal-specific = top 50 differentially upregulated genes between the fetal vs. embryonal histologies; embryonal-specific = top 50 differentially upregulated genes between the embryonal vs. fetal

histologies; mesenchymal = top 50 differentially upregulated genes between the mesenchymal histology vs. normal liver; mesenchymal-specific = top 50 differentially upregulated genes between the mesenchymal vs. embryonal histologies. Color scales indicate the average expression of the signature obtained by the AddModuleScore function in Seurat. **e** RNA velocity (using data from HB1, HB6, HB15) and pseudotime analyses (using all tumoroid data) projected onto the UMAP representation from **a**. Color scale indicates pseudotime. **f** UMAP plots showing relative expression of published gene signatures of hepatoblastoma from single-cell RNA sequencing and bulk gene expression analyses (gene lists in Supplementary Data 5). Color scales indicate average expression of the signature obtained by the AddModuleScore function in Seurat. **g** UMAP plots showing relative expression of published gene signatures of normal human hepatoblast differentiation along the hepatocyte and cholangiocyte trajectories (gene lists in Supplementary Data 5). Color scales indicate average expression of the signature obtained by the AddModuleScore function in Seurat. See also Supplementary Fig. S7, S8.

single-cell RNA sequencing of normal human fetal livers at different post-conception timepoints[51] (Fig. 5g, gene lists in Supplementary Data 5). All of the hepatoblastoma cells in the tumoroids showed high expression of hepatoblast marker genes corresponding to post-conception week (PCW) 5–6, while those cells expressing the fetal-specific gene signature showed the best correlation with the fetal hepatocyte gene signature corresponding to PCW 12-17 (Fig. 5g, gene lists in Supplementary Data 5). While the cells expressing the embryonal-specific gene signature did not clearly correspond to one

of the developmental stages of hepatocyte or cholangiocyte differentiation, it is notable that two markers of these cells, *VIM* and *SOX4*, are among those genes differentially expressed in fetal cholangiocytes corresponding to PCW 7-17 (Fig. 5g, gene lists in Supplementary Data 5). Of note, the hepatoblastoma tumoroids contained almost no cells with the adult hepatocyte or adult cholangiocyte gene signatures (Fig. 5g, gene lists in Supplementary Data 5), suggesting that the tumoroid cultures were not contaminated by normal liver cells.

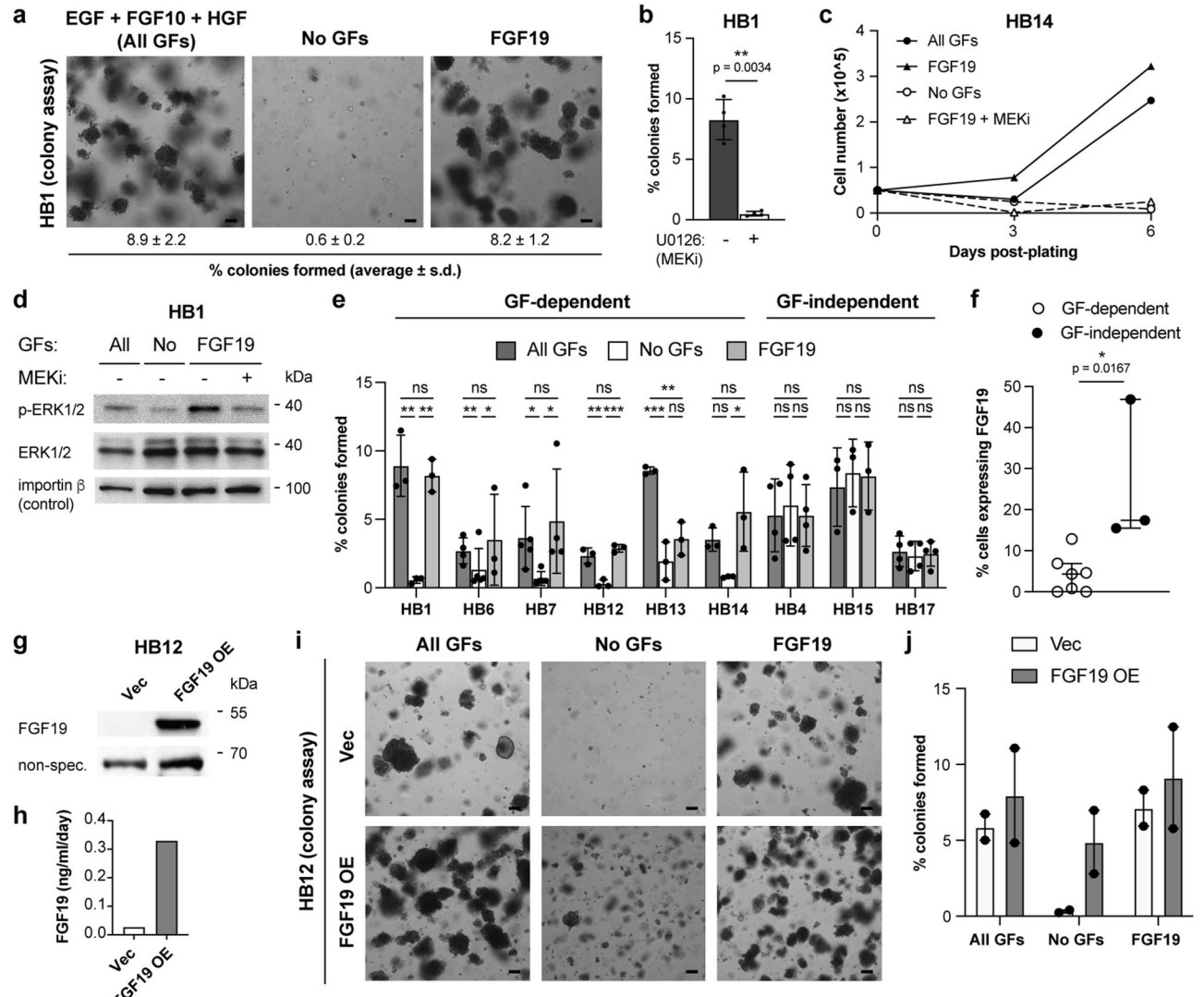

**Fig. 6 | FGF19 is sufficient to promote proliferation of FGF19-negative tumoroid cells in vitro. a** Brightfield images of FGF19-negative hepatoblastoma colonies (HB1) seeded as single cells and grown for 2 weeks in media containing EGF + FGF10 + HGF (All GFs), no GFs, or FGF19. Numbers indicate % colonies formed, mean and s.d. of $n = 3$ independent experiments. **b** Quantification of colony assay of HB1 cells grown in media with all GFs or all GFs + MEK inhibitor U0126 (MEKi, 5 μM). Mean and s.d., $n = 3$ independent experiments, paired two-tailed t-test. **c** Growth curve of HB14 tumoroids in the presence of All GFs, FGF19, no GFs, or FGF19 + U0126. Filled circles: All GFs, filled triangles: FGF19, open circles: No GFs, open triangles: FGF19 + U0126. **d** Immunoblot of p-ERK1/2, total ERK1/2, and control (importin β) in HB1 in the presence of All GFs, no GFs, FGF19, or FGF19 + U0126. **e** Quantification of colony assay of the indicated tumoroids in media containing all GFs, no GFs, or FGF19. Mean and s.d., exact p-values by ANOVA/mixed effects with uncorrected Fisher's LSD – HB1 ($n = 3$ independent experiments): 0.54 (All GFs vs.

FGF19), 0.0014 (All GFs vs. No GFs), 0.002 (No GFs vs. FGF19); HB6 ($n = 5$): 0.86, 0.0074, 0.022; HB7 ($n = 5$): 0.64, 0.012, 0.036; HB12 ($n = 5$): 0.40, 0.0004, 0.0013; HB13 ($n = 3$): 0.002, 0.0007, 0.079; HB14 ($n = 3$): 0.17, 0.088, 0.017; HB4 ($n = 3$): 0.24, 0.44, 0.088; HB15 ($n = 3$): 0.22, 0.14, 0.71; HB17 ($n = 4$): 0.53, 0.24, 0.55.
**f** Percentage of cells expressing *FGF19* by scRNAseq. Median and interquartile range, comparing GF-dependent (open circles, $n = 6$) and GF-independent (filled circles, $n = 3$) tumoroids by two-sided Mann-Whitney test. **g** Immunoblot of FGF19 and control (non-specific/GFP) in HB12 expressing lentiviral vector or FGF19.
**h** ELISA detecting FGF19 in HB12 expressing lentiviral vector or FGF19. **i** Brightfield images of HB12 colonies expressing lentiviral vector or FGF19 in media containing All GFs, no GFs, or FGF19. **j** Quantification of colony assay for HB12 as in **i**. Mean and S.E.M., $n = 2$ independent experiments. For all panels – scale bar: 100 μm, *$p < 0.05$, **$p < 0.01$, ***$p < 0.001$, ns: not significant. See also Supplementary Fig. S9, S10.

## FGF19-negative tumoroids depend on exogenous growth factors for proliferation

Since the hepatoblastoma tumoroids grew in relatively well-defined media containing recombinant growth factors, we could systematically test the requirements for these added growth factors. We dissociated tumoroids, resuspended single cells in Matrigel and media with or without exogenous growth factors, and monitored colony formation over 2 weeks. For the majority of the hepatoblastoma tumoroids (6 of 9 tested), colony formation was abolished in the absence of the standard growth factors usually included in the media (EGF, FGF10, and HGF) (Fig. 6a, e) or in the presence of a MEK inhibitor,

U0126 (Fig. 6b, Supplementary Fig. S9a). FGF19 could substitute by itself for the traditional growth factors used in the media (Fig. 6a, e). Detection of downstream phosphorylation of ERK1/2, a consequence of active Ras/MAPK signaling, showed that the removal of growth factors reduced phospho-ERK, while FGF19 promoted ERK phosphorylation to a similar level as the combination of EGF, FGF10, and HGF (Fig. 6d).

In contrast to the tumoroids that depended on exogenous growth factors for colony formation, three of the hepatoblastoma tumoroids (HB4, HB15, and HB17) were able to form colonies even in the absence of exogenous growth factors (Fig. 6e). Interestingly, growth factor

independence correlated with a higher percentage of cells expressing *FGF19* and *FGF9*, but not *FGF3* and *FGF8*, as determined by single cell RNA sequencing (Fig. 6f, Supplementary Fig. S9b-d).

In addition to the observed dependence on FGF19 or other growth factors for colony formation from single cells, we confirmed that intact tumoroids grown in the absence of growth factors failed to increase their cell number, while FGF19 was sufficient to stimulate tumoroid growth (Fig. 6c). Similar to the results observed with the addition of exogenous recombinant FGF19, overexpression of FGF19 in growth factor-dependent tumoroid cells also permitted colony formation and cell proliferation in the absence of exogenous growth factors (Fig. 6g-j, Supplementary Fig. S9e-f).

We further investigated the downstream mechanism by which exogenous growth factors exert their effects on cell proliferation. FGF19-negative tumoroids treated with MEK inhibition and withdrawal of exogenous growth factors arrested predominantly in G0/G1 phase of the cell cycle (Supplementary Fig. S10a-b, d). When the MEK inhibitor was washed out and media containing growth factors was added to the tumoroids, the cells progressed into S phase with $37 \pm 8\%$ EdU incorporation, as compared to $18 \pm 2\%$ when grown in media without growth factors ($n = 3$ independent experiments, $p = 0.049$ by two-tailed t test, Supplementary Fig. S10c-d). These results suggested that despite constitutive Wnt pathway activation, the tumor cells require additional growth factors acting through the MAPK pathway to progress from G1 into S phase.

### Endogenous *FGF19* expression in a subset of tumoroids bypasses the requirement for exogenous growth factors

Next, we focused on the hepatoblastoma tumoroids that contained a subpopulation of *FGF19*-expressing cells, which could form colonies even in the absence of exogenous growth factors (HB4, HB15, HB17 in Fig. 6e). To determine whether this independence from exogenous growth factors depended on the expression of *FGF19* by a subpopulation of tumoroid cells, we evaluated colony formation and tumoroid growth upon inhibition of FGF19 signaling. Indeed, depleting *FGF19* in HB15 using a lentiviral shRNA (Supplementary Fig. S11a-b) reduced the ability of single cells to form colonies and of intact tumoroids to grow in the absence of exogenous growth factors (Fig. 7a-b, d-e). These phenotypes were partially rescued by the presence of either exogenous FGF19 or the combination of EGF, FGF10, and HGF (Fig. 7a-b, d-e). FGF19 knockdown in both HB4 and HB17 similarly reduced colony formation in the absence of exogenous growth factors (Fig. 7c).

Consistent with the requirement for FGF19 signaling, the addition of BLU9931, a selective FGFR4 inhibitor[52], to *FGF19*-expressing tumoroid cells also reduced colony formation in the absence of exogenous growth factors, with an average IC50 of $1.24 \pm 0.31$ μM across three different tumoroids (Supplementary Fig. S11c). Inhibition of FGFR4 reduced phosphorylation of ERK, a marker of MAPK pathway activation (Supplementary Fig. S11d). In addition, treatment of *FGF19*-expressing tumoroids with the MEK inhibitor U0126, as well as a second more potent MEK inhibitor, trametinib, inhibited both colony formation and tumoroid growth in the absence of exogenous growth factors, with 50% growth inhibition at estimated concentrations of ~5-6 μM and 10-17 nM, respectively (Supplementary Fig. S11e-h). Thus, these FGF19-positive hepatoblastoma tumoroids depended on the endogenous expression of *FGF19* and subsequent signaling through FGFR4 and MAPK.

Finally, to test whether the endogenously expressed growth factors provided a paracrine effect, we investigated whether conditioned media from the FGF19-positive tumoroids, HB4 and HB15, growing in the absence of exogenous growth factors, could support the colony formation of FGF19-negative tumoroid cells. Indeed, HB1 cells, which require exogenous growth factors, formed colonies when grown in conditioned media from either HB4 or HB15 (Fig. 7f-g). Colony

formation in these conditions was inhibited by the MEK inhibitor, U0126 (Fig. 7f-g), consistent with the hypothesis that FGF19 (and/or other growth factors) secreted by HB4 and HB15 activates MAPK signaling in FGF19-negative HB1 cells.

### *FGF19* expression depends on both Wnt/β-catenin and the biliary transcription factor SOX4

We next investigated the mechanism by which *FGF19* expression is induced in hepatoblastoma cells using the FGF19-positive tumoroids. In normal tissues that express *FGF19*, transcriptional regulation occurs through the farnesoid X receptor (FXR) and FXR response elements, which are activated by bile acids[42,53,54]. In HB15 cells, *FGF19* expression was not increased by the addition of bile acids (cholic acid and chenodeoxycholic acid) or the FXR agonist cilofexor, nor reduced by an FXR antagonist, guggulsterone[55], suggesting that the regulation of *FGF19* in these cells is uncoupled from FXR (Supplementary Fig. S12a-d).

Since *FGF19* expression colocalized to a subset of cells within the embryonal components of hepatoblastoma that showed high levels of both nuclear β-catenin and Wnt target gene expression (Fig. 2b-c, e-h, Fig. 3a-c), we asked whether the upregulation of FGFs in these tumoroids depended on β-catenin. In both HB15 and HB17, either lentiviral shRNA knockdown of *CTNNB1* or overexpression of a dominant negative mutant of the transcription factor *TCF4* (DN TCF4), which blocks β-catenin-mediated transcription[56], resulted in >2-fold reduction in the expression of *FGF19* (Fig. 8a, g, Supplementary Fig. S12f). Consistent with the requirement of FGFs for the proliferation of these cells, inhibition of β-catenin mediated transcription also reduced colony formation in the absence of exogenous growth factors (Fig. 8b-c, h, Supplementary Fig. S12g-h). Colony formation was only partially rescued by the addition of recombinant FGF19 (Fig. 8b-c, h, Supplementary Fig. S12g-h), suggesting that in the absence of active Wnt signaling, FGF19 alone is insufficient to drive cell proliferation. Instead, the combination of constitutive Wnt pathway activation and Wnt/β-catenin-dependent FGF19 together allow these tumor cells to progress through the cell cycle and form colonies in the absence of exogenous growth factors.

Finally, we determined whether *FGF19* expression in HB15 and HB17 depends on the biliary transcriptional program, as *FGF19* expression was found in a subset of the *SOX4*-expressing embryonal hepatoblastoma cells with cholangiocytic features (Figs. 2b-c, e-h, 3d-e). Depletion of *SOX4* by lentiviral shRNA in both tumoroids significantly reduced the expression of *FGF19* (Fig. 8d, g). The reduction in *FGF19* expression was proportional to the level of *SOX4* knockdown (Supplementary Fig. S12e). Conversely, overexpression of *SOX4* in HB15 reproducibly resulted in upregulation of *FGF19* as well as the previously reported SOX4 target gene *TEAD2* (Fig. 8j). *SOX4* depletion reduced colony formation both in the absence and presence of growth factors, suggesting that SOX4 promotes colony formation through induction of *FGF* expression as well as an FGF-independent mechanism (Fig. 8e-f, i). Depletion of *SOX4* did not consistently affect the expression of Wnt target genes (Fig. 8d). Similarly, knockdown of *CTNNB1* did not significantly affect the expression of *SOX4* or its targets, suggesting that *SOX4* is not downstream of Wnt/β-catenin (Supplementary Fig. S12f). Overall, these results suggest that *SOX4* cooperates with stabilized β-catenin to induce the expression of *FGF19* but not other canonical Wnt target genes.

## Discussion

In summary, we show in a pediatric cancer with an intrinsically low mutational burden that transcriptional heterogeneity driven by developmental lineage programs can provide a mechanism of generating cell diversity that benefits the tumor. While nearly all hepatoblastomas contain somatic activating mutations in *CTNNB1*, our study, as well as that of another group[57], shows that the transcriptional output

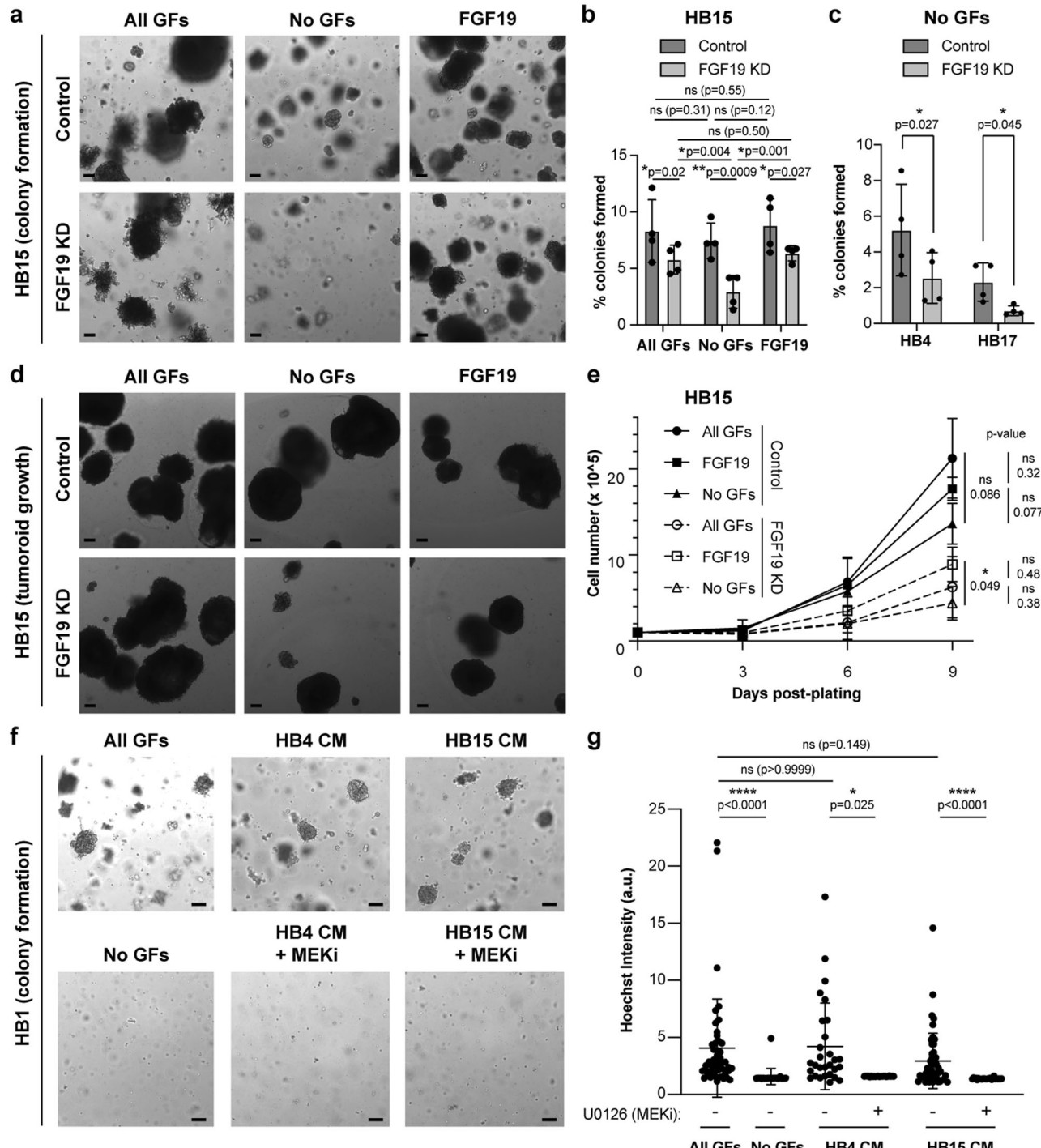

**Fig. 7 | Endogenous *FGF19* expression by a subset of tumor cells bypasses the requirement for exogenous growth factors. a** Brightfield images of HB15 colonies expressing lentiviral shRNA to luciferase (control) or *FGF19* seeded as single cells after 3 days of antibiotic selection and grown for 2 weeks in media containing EGF + FGF10 + HGF (All GFs), no GFs, or FGF19. **b** Quantification of colony assay as in **b**. Mean and s.d., *n* = 4 independent experiments, 2-way ANOVA with uncorrected Fisher's LSD. **c** Quantification of colony formation assays of HB4 and HB17, expressing lentiviral shRNA to luciferase (control) or *FGF19*, seeded as single cells and grown for 2 weeks in media containing no GFs. Mean and s.d, *n* = 4 independent experiments, paired two-tailed t-test. **d** Brightfield images of HB15 tumoroids expressing lentiviral shRNA to luciferase (control) or *FGF19*, at 9 days after initially seeding 100,000 cells as tumoroids, in media containing All GFs, no GFs, or FGF19. **e** Growth curve of experiment depicted in **d**. Mean and s.d., *n* = 3 independent experiments, 2-way ANOVA with uncorrected Fisher's LSD. Filled circles: control +

All GFs, filled squares: control + FGF19, filled triangles: control + No GFs, open circles: FGF19 KD + All GFs, open squares: FGF19 KD + FGF19, open triangles: FGF19 KD + No GFs. **f** Brightfield images of HB1 seeded as single cells and grown for 2 weeks in media containing All GFs, no GFs, HB4 conditioned media (CM) without or with MEK inhibitor (MEKi, U0126 5 μM), and HB15 conditioned media (CM) without or with MEK inhibitor (MEKi, U0126 5 μM). **g** Quantification of Hoechst intensities of tumoroid colonies from **f**. Mean and s.d., *n* = 46, 24, 29, 22, 49, and 25 tumoroids for each condition in **f**, respectively, Kruskal-Wallis with Dunn's multiple comparisons test. Similar results obtained for *n* = 2 independent experiments. Due to the inability to accurately control the amount of growth factor secreted into conditioned media by a given tumoroid line across multiple experiments, data from each experiment was compared separately and results from one experiment are shown. For all panels – scale bar: 100 μm, *: *p* < 0.05, **: *p* < 0.01, ****: *p* < 0.0001, ns: not significant. See also Supplementary Fig. S11.

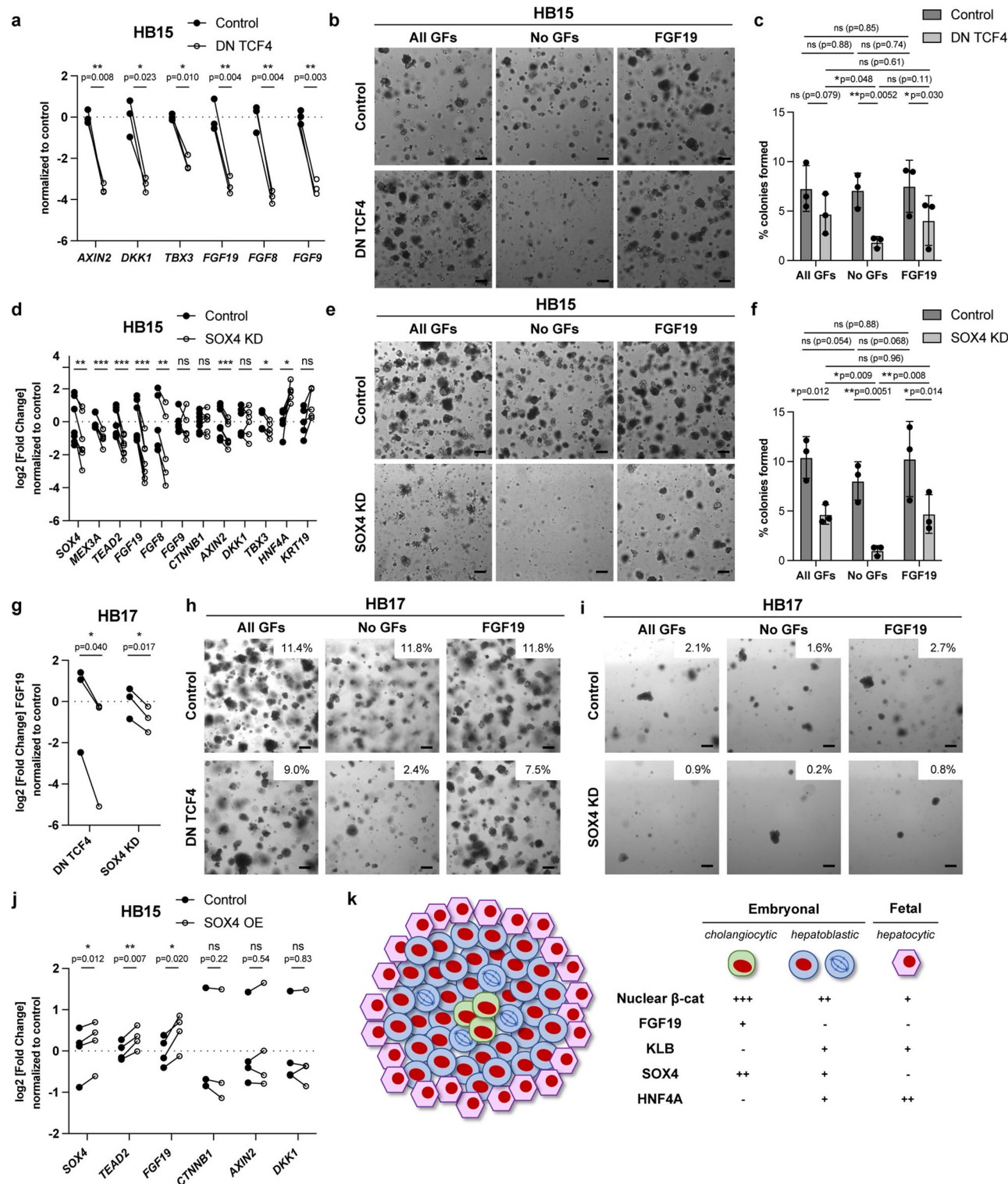

of Wnt pathway activation is modulated within individual tumors and exists along a gradient across the different histologies, with implications for their biologic behavior. We confirm lower expression of Wnt target genes and lower levels of nuclear β-catenin within fetal regions of hepatoblastoma compared to the embryonal regions, as has been previously reported[18,19]. Within the embryonal histology, we find regions of particularly high Wnt target gene expression and high nuclear β-catenin levels corresponding to high expression of cholangiocyte markers and a low proliferative index, surrounded by a zone of intermediate Wnt target gene expression and nuclear β-catenin,

with decreased cholangiocytic markers and increased proliferation. In the *SOX4*-expressing components of the tumor, a subset of cells with higher nuclear β-catenin levels and cholangiocytic markers express *FGF19*, which promotes proliferation of surrounding *HNF4A*-expressing hepatoblastic cells (Fig. 8k).

We favor a model whereby SOX4 and nuclear β-catenin cooperate to induce the expression of *FGF19* through binding at its promoter. This is based on a significant body of prior work suggesting that SOX transcription factors interact with β-catenin in transcriptional complexes to modulate the outcome of Wnt activation[58–62]. Nonetheless,

**Fig. 8 | Endogenous *FGF19* expression in tumor cells depends on β-catenin and SOX4. a** qRT-PCR of HB15 expressing lentiviral *GFP* (control, filled circles) or dominant negative *TCF4* (DN TCF4, open circles). $n = 3$ independent experiments, paired two-tailed t-test. **b** Brightfield images of HB15 colonies expressing lentiviral *GFP* or DN *TCF4* in media containing EGF + FGF10 + HGF (All GFs), no GFs, or FGF19. **c** Quantification of colony assay in **b**. Mean and s.d., $n = 3$ independent experiments, ANOVA, uncorrected Fisher's LSD. **d** qRT-PCR of HB15 expressing lentiviral shRNA to luciferase (control, filled circles) or *SOX4* (open circles). Exact p-values, paired two-tailed t-test – *SOX4* ($n = 7$ independent experiments): 0.0001, *MEX3A* ($n = 6$): 0.0005, *TEAD2* ($n = 7$): 0.0001, *FGF19* ($n = 7$): 0.0001, *FGF8* ($n = 5$): 0.0043, *FGF9* ($n = 5$): 0.58, *CTNNB1* ($n = 7$): 0.48, *AXIN2* ($n = 7$): 0.0001, *DKK1* ($n = 6$): 0.88, *TBX3* ($n = 5$): 0.019, *HNF4A* ($n = 5$): 0.028, *KRT19* ($n = 5$): 0.066. **e** Brightfield images of HB15 colonies expressing lentiviral shRNAs to luciferase (control) or *SOX4* in media containing All GFs, no GFs, or FGF19. **f** Quantification of colony assay as in **e**. Mean and s.d., $n = 3$ independent experiments, ANOVA, uncorrected Fisher's LSD. **g** qRT-PCR for *FGF19* in HB17 expressing lentiviral control (luciferase shRNA or *GFP*, filled circles), compared to DN TCF4 or *SOX4* shRNA, respectively (opens circles). $n = 3$ independent experiments each, paired two-tailed t-test. **h** Brightfield images of HB17 colonies expressing lentiviral *GFP* or DN TCF4, in media containing All GFs, no GFs, or FGF19. % colonies formed indicated. **i** Brightfield images of HB17 colonies expressing lentiviral shRNA to luciferase (control) or *SOX4* in media containing All GFs, no GFs, or FGF19. % colonies formed indicated. **j** qRT-PCR of HB15 expressing lentiviral GFP (control, filled circles) or *SOX4* (open circles). $n = 4$ independent experiments, paired two-tailed t-test. k) Model depicting the spatial organization of different tumor cell types in hepatoblastoma and relative levels of nuclear β-catenin, FGF19, KLB, SOX4, and HNF4A. For all panels – scale bar: 200 μm, *: $p < 0.05$, **: $p < 0.01$, ***: $p < 0.001$, ns: not significant. Source data are provided as a Source Data file. See also Supplementary Fig. S12.

our experiments have not formally excluded the alternative model that SOX4 and β-catenin indirectly promote expression of *FGF19* through other downstream transcription factors or that *FGF19*-expressing cells uniquely require SOX4 and β-catenin for survival, explaining why depleting these genes reduces expression of *FGF19*. Finally, the mechanism by which higher levels of nuclear β-catenin are induced in the subset of *SOX4*[+] cells that express *FGF19* remains to be determined.

Our work has uncovered a tumor-intrinsic source of growth factors that leads to increased proliferation within the embryonal components of hepatoblastoma, which is key to their aggressive biology. Although the expression of *FGF19* and *FGF8* have previously been reported in hepatoblastoma cell lines and primary tumors[24,25], our study now ascribes their expression to a subset of cells with embryonal histology that show cholangiocytic differentiation. Importantly, we have localized *FGF*-expressing cells spatially within primary hepatoblastoma tissues, where they form distinctive foci in regions of high proliferative activity, resembling morphogen-secreting signaling centers in embryonic development. Our results indicate that activated Wnt/β-catenin is insufficient for tumor cell cycle progression and that FGFs or other growth factors are required to promote the transition from G1 to S phase. Of note, genomic amplification of *FGF19* has previously been identified in hepatocellular carcinoma[63,64] and more rarely in hepatoblastoma[22], supporting the notion that FGF19 signaling can provide a growth advantage in liver cancers.

Our finding that hepatoblastoma cells require exogenous growth factors despite constitutive activation of the Wnt pathway indicates that these cells are not intrinsically self-sufficient. Although embryonal hepatoblastoma cells can bypass the requirement for exogenous growth factors through the expression of *FGF*s, including *FGF19*, our studies suggest that a significant portion of hepatoblastoma cells do not proliferate in the absence of growth factors from cell-extrinsic sources. These findings have significant implications for tumorigenesis in infancy and childhood, as hormones and other signals that promote organismal growth may also influence the proliferation of incipient cancer cells. Given that FGF19 is secreted by the intestine in response to feeding and bile acid secretion[30], our studies warrant further investigation as to how endocrine signals and environmental modifications such as diet may promote the growth of hepatoblastoma. Our compendium of primary hepatoblastoma tumoroids provides a powerful platform to further elucidate the effects of different growth factors and nutrient-signaling pathways.

Since hepatoblastoma is a rare cancer, with an incidence of only ~100 cases per year in the US, our study is limited in sample size by the availability of specimens at a single institution. Nonetheless, our cohort shows that a significant proportion of primary hepatoblastomas exhibit *FGF19* expression within the embryonal components. Furthermore, FGF19 plays an important proliferative role in those patient-derived tumoroids that contain *FGF19*-expressing cells, while tumoroids that do not contain *FGF19*-expressing cells require

exogenous growth factors. We note that not all the tumors or tumoroids we analyzed grow by the patterns that we delineate, but this is not unexpected given that these are cancers. We believe the combination of our careful histology-guided and spatial transcriptomic studies of primary hepatoblastoma specimens, in addition to single cell analyses and functional studies of patient-derived tumoroids, provide complementary data in a substantial, representative cohort of patient samples that support our conclusions. Given the rarity of this disease, further validation with a larger patient cohort will require collaboration among consortia involving multiple institutions.

Overall, our work establishes that hepatobiliary developmental lineage programs modulate the transcriptional outcome of activated Wnt signaling in hepatoblastoma, generating intratumoral heterogeneity that provides a growth advantage. Our results support a cooperative model of tumor growth whereby cholangiocytic tumor cells promote the proliferation of surrounding tumor cells expressing hepatic markers, linking cell state and function. These results imply that tumor growth is facilitated when a Wnt pathway mutation occurs in a cell that can give rise to daughter cells that adopt both cholangiocytic and hepatic fates. The relative tumor-initiating capacity of the different hepatoblastoma cell types individually or in combination remains to be tested and likely depends not only on their ability to respond to proliferative signals but also on the level of plasticity between the cell states.

## Methods

### Ethical regulations

The research herein complies with all applicable institutional and federal ethical regulations. The study protocols for human subjects research was approved by the Stanford University Institutional Review Board. For all tumoroids described in this study, informed consent was obtained from the patient's legal guardian(s) for tissue banking and publication of results. Informed consent for research on archival specimens previously obtained by the Department of Pathology was exempted by the IRB.

### Human tissues

Archival hepatoblastoma specimens from 2003 to 2017 were identified through the Stanford Department of Pathology. Human hepatoblastoma specimens from patients diagnosed at Lucile Packard Children's Hospital between 2018 and 2023 were collected at the time of biopsy or resection and processed as described below. All patient-derived tissues and cells described in this study were obtained at Stanford/Lucile Packard Children's Hospital except for HB7, which was a gift from Bruce Wang (UCSF)[48].

### Laser capture microdissection & Smart-3SEQ

Hematoxylin and eosin-stained sections of archival hepatoblastoma specimens were reviewed by a board-certified pediatric pathologist

(F.K.H.), and regions of fetal, embryonal, and mesenchymal histology, as well as normal liver were marked. Smart-3SEQ[33] was performed as follows: Briefly, fresh sections of each FFPE block were cut on a microtome at 7 μm thickness and mounted on glass slides with polyethylene naphthalate membranes (Thermo Fisher Scientific, LCM0522) and stored in a nitrogen chamber. In preparation for laser capture microdissection, slides were immersed sequentially for 20 seconds each in xylenes (three times), 100% ethanol (three times), 95% ethanol (two times), 70% ethanol (two times), water, hematoxylin (Dako S3309), water, bluing reagent (Thermo Fisher Scientific 7301), water, 70% ethanol (two times), 95% ethanol (two times), 100% ethanol (three times), and xylenes (three times). Immediately after staining, cells were dissected on an ArcturusXT LCM System using the ultraviolet (UV) laser to cut the sample and the infrared laser to adhere it to a CapSure HS LCM Cap (Thermo Fisher Scientific, LCM0215). Roughly 500 cells were captured by area, based on density estimates by cell counting on small areas. After LCM, the cap was sealed in a 0.5 mL tube (Thermo Fisher Scientific, N8010611) and stored at −80 °C until library preparation. Sequencing libraries were prepared using the Smart-3SEQ protocol (https://genome.cshlp.org/content/suppl/2019/10/24/gr.234807.118.DC1/Supplemental_File_2.pdf) for FFPE tissue on an Arcturus LCM HS cap, with the pre-SPRI pooling option and a single batch for all samples. Libraries were characterized immediately and stored at −20 °C until sequencing. Sequencing was performed on the pooled library using the NextSeq 500/550 High Output V2 kit with 75 cycles (read 1: 76 cycles, index 1: 2 bp, index 2: 8 bp). Reads were aligned to the human reference genome with STAR and read counts were obtained using featureCounts (Subread). PCA analysis was performed on results from 72 samples (excluding 2 outliers) using FactoMiner (version 2.8) on the top 1500 expressed genes. Clustering was performed with Cluster 3.0 and Java TreeView using the following parameters: filtering for SD > 2.5 (1222 genes), Pearson correlation (uncentered), complete linkage. Differential expression analyses were performed with DESeq2[34] (version 1.38.3), excluding 2 outlier samples and collapsing repeated samples. Total transcript counts were normalized for sequencing depth and RNA content using the median of ratios in DESeq2. Differential expression was performed with the Wald test adjusted for multiple hypothesis testing with the Benjamini-Hochberg procedure, to obtain padj values that represented the false discovery rate (FDR). The GSEA 4.3.2 application was used to run GSEAPreranked on gene lists obtained by DESeq2 ranked by padj, the gene set "hallmark.all.v2023.1.Hs.symbols.gmt" from MSigDB, number of permutations = 1000. A cut-off of FDR < 0.05 was used to identify the most significantly enriched pathways.

## Spatial transcriptomics of hepatoblastoma sections
Manufacturer's instructions were followed for spatial transcriptomics using the Visium platform (10x Genomics). Briefly, 5 μm thick FFPE sections were collected on the Visium slides (10x Genomics), de-paraffinized, stained with Hematoxylin and Eosin prior to imaging on Keyence BZX 800 microscope (Keyence corp). Slides were subjected to de-crosslinking followed by probe hybridization overnight. This was followed by probe ligation, extension and elution. Ligated probes were subjected to index PCR to generate sequencing-ready libraries. The number of amplification cycles for index PCR were determined by qPCR (QuantStudio 12 K Flex, Thermo Scientific). Libraries were quantified on a BioAnalyzer 2100 instrument (Agilent), pooled and sequenced on a NextSeq 500 instrument (Illumina) using the manufacturer-recommended read depth and read length. Space Ranger software (10x Genomics) was used for downstream analysis and the Loupe Browser software (10x Genomics) was used for UMAP clustering and visualization.

Clusters were assigned to fetal, embryonal, or mesenchymal based on the overlap of cluster marker genes with the differentially expressed gene lists identified by Smart-3SEQ (listed in Supplementary Data 5: Wu_Gene_Sigs). Spots with FGF19 expression were identified. Pie charts were generated to show the distribution of spots per cluster for all spots and specifically for the FGF19+ spots. The percentage of FGF19+ spots / total spots were calculated for each cluster. Chi-squared tests were performed to determine the statistical significance of the relationship between the FGF19+ and the embryonal spots (cluster 4 in HB4, cluster 1 in HB17).

## Tissue processing
Fresh tumor and normal liver tissue were fixed overnight at room temperature in 10% neutral buffered formalin, dehydrated, cleared with HistoClear (Natural Diagnostics), and embedded in paraffin. Sections were cut at 5 μm thickness, rehydrated, and processed for hematoxylin/eosin staining, immunofluorescence, or in situ hybridization as described below.

## Immunofluorescence of FFPE sections
Following deparaffinization and hydration, antigen retrieval of FFPE tissue sections was performed with Tris buffer at pH 8.0 (Vector Labs H-3301) in a pressure cooker for 20 minutes. Slides were permeabilized with phosphate-buffered saline (PBS) containing 0.1% Tween-20, then blocked in 5% normal donkey serum in PBS containing 0.1% Triton-X. Sections were incubated with primary antibody, washed three times with PBS, then incubated with secondary antibody. The stained sections were washed with PBS and mounted in Prolong Gold with DAPI (Invitrogen). The following antibodies were used: HNF4α (rabbit polyclonal, 1:50; Santa Cruz sc8987, H-171), KRT19 (rabbit, 1:100, Abbomax 602-670, clone nan), MKI67 (rat monoclonal, 1:100; eBioscience 14-5698-82, clone SolA15), phospho-histone H3 (ser10) (rabbit polyclonal, 1:1000, Millipore 06-570), β-catenin (mouse FITC-conjugated, 1:50, BD β-Catenin clone #14, custom AB #624044; detected with secondary antibody to FITC). Images were captured on a Zeiss Axio Imager Z.2.

## RNAscope in situ hybridization
Single and dual in situ hybridizations were performed using the manual RNAscope 2.5 HD Assay-Red Kit and RNAscope 2.5 HD Duplex Assay Kit (Advanced Cell Diagnostics), respectively, according to the manufacturer's instructions. Images were taken at 20x magnification on a Zeiss Axio Imager Z.2. Probes used in this study were Axin2 (target region: 502–1674), FGF19 (target region: 457–2,128), KLB (target region: 729–1680), SOX4 (target region: 5–3238).

## Culture of hepatoblastoma tumoroids
Tumor tissue was collected from patients with hepatoblastoma at the time of biopsy, tumor resection, or liver transplant and immediately processed[44]. Briefly, tissue was placed immediately in cold Advanced DMEM/F12 with 1x GlutaMAX (Gibco 35050061), 10 mM HEPES, and 100 U/mL Penicillin/Streptomycin (Gibco 15140122) and kept at 4 °C. Tissue was minced with a razor blade, washed with DMEM (high glucose, with GlutaMAX), containing 1% FBS (Omega Scientific Inc. FB-01) and 100 U/mL Penicillin/Streptomycin (Gibco 15140122) and centrifuged at 150 g x 5 min. Wash media was removed. Cell clusters were then embedded in 20 μL Matrigel (Growth Factor Reduced, Phenol Red-free, LDEV-free, Corning 356231 or Cultrex Reduced Growth Factor Basement Membrane Extract, Type 2, Bio-techne/R&D Systems 3533-005-02) and layered with media comprised of 100 U/mL Penicillin/Streptomycin (Gibco 15140122), 1x GlutaMAX (Gibco 35050061), 10 mM HEPES, 1x B27 supplement (Gibco 17504044), 1x N2 supplement (Gibco 17502048), 1.25mM N-acetyl cysteine (Sigma-Aldrich A9165), 10 mM nicotinamide (Sigma-Aldrich N3376), 10 mM Y27632 (PeproTech 1293823), 5 μM A83-01 (Tocris 2939), 3 nM dexamethasone (Tocris 1126), 50 ng/ml recombinant human EGF (PeproTech AF-100-15), 100 ng/ml recombinant human FGF10 (PeproTech 100-26), 25 ng/ml recombinant human HGF (PeproTech 100-39H),

0.1 mg/mL Normocin (Invivogen ant-nr-2), in Advanced DMEM/F12 (Gibco 12634010, with glucose, non-essential amino acids, sodium pyruvate, and phenol red).

Tumoroids were passaged, between 1:4 to 1:10, every 1-2 weeks. Cell clusters were released from Matrigel by incubation in dispase 5 U/mL (Stem Cell Technologies 07913) for up to 30 minutes at 37 °C, washed twice with liver perfusion media containing 5% FBS and once with William's media containing 5% FBS. FBS was included to prevent tumoroids from sticking to pipettes and other plasticware. Tumoroids were centrifuged at 50 g x 5 min after each wash. All wash media was removed, and tumoroids were re-embedded in 20 μL Matrigel drops (8 drops per well of a 6-well plate) and overlayed with tumoroid media as above.

In situations where single cells were required for either cell counting or colony formation assays, tumoroids released from Matrigel were washed as above, then incubated in TrypLE for 15 minutes at 37 °C. When isolating tumoroids or single cells, suspensions were centrifuged at 50 g or 300 g, respectively, for 5 minutes.

### Tumoroid histology
Tumoroids were released from Matrigel as described above, fixed in 4% paraformaldehyde at room temperature for 1 hour, washed twice in PBS. Tumoroids were resuspended in Histogel (Thermo Scientific) and distributed into a CHEF plug mold. Plugs containing tumoroids in Histogel were fixed, dehydrated, cleared, and embedded in paraffin identically to primary tissues as above.

### Tumoroid growth analysis
Tumoroids were released from Matrigel using dispase (30 minutes at 37 °C) and a fraction of the tumoroids were incubated in TrypLE (15 minutes at 37 °C) to obtain single cells for cell counting. Single cells were counted using the Cellomics cell counter. Intact tumoroids were seeded at a density of 50,000 or 100,000 cells per well in a 6-well plate (either 4 or 8 drops per well of 20 μL Matrigel containing tumoroids). Media was changed every 3-4 days and cells in each well were counted at desired timepoints.

### DNA isolation and *CTNNB1* sequencing
DNA was extracted using standard phenol/chloroform or the Qiagen AllPrep DNA/RNA Kit (Qiagen, Hilden, Germany). *CTNNB1* PCR for a 1046 bp fragment encompassing exon 2 through exon 4 was performed with the following primers:

CTNNB1 ex2-F 5′-AGCGTGGACAATGGCTACTCAA
CTNNB1 ex4-R: 5′-ACCTGGTCCTCGTCATTTAGCAGT

PCR was performed with an annealing temperature of 70 °C, extension time of 1 minute, and 35 cycles. Products were isolated by gel extraction and analyzed by Sanger sequencing with the same primers above.

### RNA isolation and qRT-PCR
Total RNA was extracted using the Qiagen RNAeasy Mini Kit (Qiagen, Hilden, Germany) and reverse transcribed (High Capacity cDNA Reverse Transcription Kit; Life Technologies, Carlsbad, CA) according to the manufacturer's protocol. Quantitative RT-PCR was performed with TaqMan Gene Expression Assays (Applied Biosystem, Waltham, MA) on an StepOnePlus Real-Time PCR System (Applied Biosystems, Waltham, MA). Relative target gene expression levels were calculated using the delta-delta CT method. Gene Expression Assays used were: *GAPDH* (Hs99999905_m1, FAM-MGB), *AXIN2* (Hs00610344_m1, FAM-MGB), *CTNNB1* (Hs00355045_m1, FAM-MGB), *DKK1* (Hs00183740_m1, FAM-MGB), *TBX3* (Hs00195612_m1, FAM-MGB), *FGF19* (Hs00192780_m1, FAM-MGB), *FGF8* (Hs00171832_m1, FAM-MGB), *FGF9* (Hs00181829_m1, FAM-MGB), *KRT19* (Hs01051611_gH, FAM-MGB), *HNF4A* (Hs00230853_m1, FAM-MGB), *SOX4* (Hs00268388_s1, FAM-MGB),

*MEX3A* (Hs00863536_m1, FAM-MGB), and *TEAD2* (Hs01055894_m1, FAM-MGB), all from Thermo Fisher Scientific (Waltham, MA).

### FGF19 ELISA
Intact tumoroids were embedded in Matrigel at a density of 50,000 or 100,000 cells per well in a 6 well plate in media lacking FGF19. Media was collected for ELISA between 24-72 hours after plating. Sandwich ELISA was performed using the human FGF19 ELISA kit (Invitrogen EHFGF19), according to the manufacturer's instructions. Calculations of FGF19 concentration were normalized for a number of days in culture and a number of cells initially plated.

### Immunofluorescence of intact fixed tumoroids
Matrigel droplets containing intact tumoroids were seeded in 8-chamber well plates and grown in tumoroid media until the appropriate density was reached. Fixation was performed using 4% PFA at room temperature for 30 minutes, with one PBS wash performed at 10 minutes. Permeabilization, primary and secondary antibody incubations, and washes, were performed as above for FFPE sections. Imaging was performed on a spinning disk confocal, taking optical sections through a 30-micron Z-stack.

### Single-cell RNA sequencing
Tumoroids at P1 or P2 were released from Matrigel using dispase 5 U/mL (Stem Cell Technologies 07913), followed by dissociation to single cells with TrypLE Express (Gibco 12605010) and resuspension in William's media with 10% FBS. Single-cell RNA sequencing was performed using the 10x Chromium V2 (HB1) or V3 systems, targeting 5000 cells for each sample. Libraries for HB1, HB4, HB6, and HB7 were sequenced with the Illumina NovaSeq 6000 instrument with NovaSeq S1 v.1.5 Reagent Kits with the following reads: 28 bases Read 1 (cell barcode and unique molecular identifier [UMI]), 8 bases i7 Index 1 (sample index), and 91 bases Read 2 (transcript). Libraries for HB12, HB13, HB14, HB15, HB16, HB17 were sequenced with the Illumina NextSeq 550 High Output instrument with the following reads: 28 bases Read 1 (cell barcode and unique molecular identifier [UMI]), 10 bases i7 Index 1 (sample index), 10 bases i5 Index 2 (sample index), and 90 bases Read 2 (transcript). Demultiplexed fastq files were processed for single-cell counting and reference genome mapping using the Cell Ranger Software (version 3.0.2, GRCh38.3.0.0 ref genome). Dimensionality reduction by principal component analysis (PCA), graph-based clustering, and UMAP visualization were performed using Seurat (version 4.3.0). In brief, data was filtered to include only genes detected in > 3 cells, cells with > 200 and <10,000 detected genes, and cells with <15% mitochondrial genes. Data from 10 samples were normalized, combined, and integrated using canonical correlation analysis (CCA) to identify anchors. Integrated data was scaled, regressing out the read depth, percent mitochondrial genes, and percent ribosomal genes. Graph-based clustering and UMAP visualization were performed with the first 30 principal components and a resolution of 0.4. Clustering and UMAP visualization were also performed on each of the samples separately without integration, using the same parameters.

Marker genes for each cluster were identified using the FindAllMarkers function on the original RNA data (not the Integrated data). Expression levels of specific genes were plotted using the FeaturePlot function, where Feature counts for each cell are divided by the total counts for that cell and multiplied by the scale.factor, then natural-log transformed. Gene signature scores were obtained using the AddModuleScore function, using the gene lists in Supplementary Data 4. AddModuleScore calculates the average expression levels of each signature on a single cell level, subtracted by the aggregated expression of control feature sets, which are randomly selected from bins that contain all analyzed features separated by averaged expression.

RNA velocity analysis was performed by using velocyto in R to generate loom files, which were then processed by scvelo in Python. Pseudotime analysis was performed using Monocle in R.

## Protein extraction and immunoblotting

Cell pellets were incubated for 15 minutes in ice-cold RIPA buffer (50 mM Tris pH 8.0, 150 mM NaCl, 1% NP-40, 0.1% SDS, 0.5% Na deoxycholate) with freshly added protease inhibitors (Complete mini, EDTA-free, Roche), phosphatase inhibitors (PhosSTOP, Roche), 1 mM PMSF, and 1 mM DTT, at a concentration of 10^7 cells/mL. Lysates were centrifuged at max speed for 15 minutes at 4 °C and the protein-containing supernatant was stored at -80 °C until use. Protein concentration was measured using the DC Protein Assay (Bio-Rad). Lysates were diluted 1:1 in 2x Laemmli buffer, boiled, and 20 µg total protein was loaded for SDS-PAGE (miniPROTEAN TGX pre-cast gel, Bio-Rad). After transfer to a nitrocellulose membrane and blocking in 5% milk/PBST, the following antibodies were used (diluted in 5% milk/PBST unless otherwise indicated): FGF19 (mouse monoclonal, 1 µg/ml, R&D MAB969), GFP (rabbit polyclonal, 1:5000, OriGene TA150122), phospho-ERK1/2 (rabbit polyclonal, 1:1000 in 1% BSA/TBST, Cell Signaling Technology 9101S), ERK1/2 (rabbit polyclonal, 1:1000, Cell Signaling Technology 9102S), γ-tubulin (mouse monoclonal, 1:10,000, Abcam GTU-88 ab-11316), importin β1 (rabbit monoclonal, 1:2000, Cell Signaling Technology 60769S). Membranes were washed, incubated with the appropriate secondary antibody conjugated with horse radish peroxidase (HRP) and diluted in 5% milk/PBST, and detected by chemiluminescence (Western Lightning Plus ECL, ThermoFisher Scientific).

## Colony formation assay

Cells were seeded at a density of 3000 single cells per 20 µL drop of Matrigel and 8 drops per well of a 6-well plate. Media was changed every 3-4 days and colonies were assessed after 2 weeks. After removal of media, colonies in Matrigel drops were incubated for 1 hour with Hoechst 33342 (10 µg/mL) in dispase (5 U/mL). Colonies were released from Matrigel by gentle pipetting, washed once with liver perfusion media with 5% FBS, resuspended in the same media and transferred to a 96-well plate (Corning) for imaging. Colonies were imaged with the CellInsight CX7 High-Content Screening (HCS) Platform (Thermo Fisher Scientific) using bright-field, confocal Z-stack, and widefield channels, and quantified using the HCS Studio Cell Analysis Software (Thermo Fisher). Cell debris or dead cells were excluded based on colony size, nuclear intensity, and length-to-width ratio, with the same threshold applied for colony counting in all experiments.

## Cell cycle synchronization and flow cytometry of tumoroid cells

Tumoroids were incubated in media with no growth factors and U0126 5 µM for 24 hours. Media was removed and tumoroids were washed twice with media without growth factors or drug and replaced with either media containing growth factors or media without growth factors. After 35 hours, EdU 10 µM was added for 1 hour, then tumoroids were isolated from Matrigel using dispase and further dissociated to single cells using TrypLE. Single cells were washed with liver perfusion media with 5% FBS and fixed with 70% ethanol. Cells were stained with the Click-iT Plus EdU Alexa Fluor 647 Flow Cytometry Assay Kit (Thermo Fisher C10340) and propidium iodide for DNA content. Flow cytometry was performed on a BD FACS Aria II machine, using the BD FACSDiva Software.

## Lentiviral transduction of tumoroids

At 24 hours prior to transfection, 293 T cells (ATCC, CRL-3216) were plated in 6 well plates at 8 ×10^5 cells / well with DMEM (high glucose, +Glutamax, no sodium pyruvate) / 10% FBS / Pen-Strep (Gibco). Cells growing at ~70-80% confluency were transfected with 1 µg of the desired expression plasmid (Sigma Aldrich) along with 2nd generation lentiviral packaging and envelope plasmids, 0.75 µg of psPAX2 and 0.25 µg of pMD2.g (gifts from D. Trono, Addgene #12260, #12259). At 36 hrs post transfection, the media containing lentiviral particles was collected and passed through a 0.45 µm filter. Polybrene was added to a final concentration of 4 µg/mL. The filtered media containing lentivirus was added at a 1:1 dilution to target hepatoblastoma tumoroids suspended in media. A total of 4 infections, each spaced 12 hours apart, were performed. Tumoroids were transferred to antibiotic selection 24 hours after the last infection. For puromycin-selectable vectors, tumoroids were cultured in media containing puromycin 2 µg/ml for 3 days prior to isolation of cells for RNA or protein extraction or seeding for colony assays.

Predesigned shRNA in the puromycin-selectable pLKO.1 vector were obtained from Sigma Aldrich, targeting the following sequences –
*CTNNB1*: 5′-GCTTGGAATGAGACTGCTGAT (TRCN0000003845)
*FGF19*: 5′-GCTTTCTTCCACTCTCTCATT (TRCN0000040260)
*SOX4*: 5′-AGCGACAAGATCCCTTTCATT (TRCN0000018214)
Luciferase: 5′-CGCTGAGTACTTCGAAATGTC (SHC007)

Other plasmids used in the study were: pLKO.1-puro-CMV-TurboGFP (Sigma-Aldrich, SHC003), pLenti-C-mGFP-P2A-puro-FGF19 (Origene, RC203750L4), EdTP (dominant negative Tcf4, Addgene #24311), pWPXL-SOX4 (gift from Robert Weinberg, Addgene #36984).

## Reporting summary

Further information on research design is available in the Nature Portfolio Reporting Summary linked to this article.

## Data availability

The metadata and transcript counts from Smart-3SEQ generated in this study are provided in Supplementary Data 1. Raw sequencing data from the Smart-3SEQ experiment in this study are available in the Gene Expression Omnibus (GEO) under accession code GSE279385. Raw sequencing data from spatial transcriptomic and single cell RNA sequencing experiments in this study are available in GEO under accession code GSE249965 and GSE233923, respectively. Gene lists for specific signatures identified in this study and by prior single cell RNA sequencing[15,22,48–50] and microarray studies on hepatoblastoma[19] as well as single cell RNA sequencing of human fetal liver[51], all of which were used in this study, are provided in the Supplementary Data 5. Hepatoblastoma tumoroids are available upon request. Source data are provided as a Source Data file with this paper. Further information and requests for resources and reagents should be addressed to Peng V. Wu (Peng.Wu@cchmc.org). Source data are provided with this paper.

## Code availability

No custom algorithms were used in this study. Sample code for Smart-3SEQ differential gene expression analyses and 10x single cell RNA sequencing analyses are available at github.com/pengvwu/Hepatoblastoma.

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

## Acknowledgements

We are grateful to the patients and families who consented to banking tissue for research. We thank the Bass Center Tissue Biorepository and members of the Pediatric Hematology/Oncology, Liver Transplant Surgery, Pediatric Interventional Radiology, and Pathology teams at Lucile Packard Children's Hospital/Stanford Health Care for assistance with tissue procurement. We thank Bruce Wang (UCSF) for the gift of HB7. We acknowledge members of Stanford Genomics and the Chan Zuckerberg Biohub for sequencing support. We further thank the following individuals: Shirley Kwok and Pauline Chu from the Stanford Department of Pathology for tissue sectioning and slide preparation; Shahrzad Talebian from Stanford Genomics for help with 10x Visium library preparation; Catriona Logan, D. Berfin Azizoglu, Xin Wang, Ellen Rim, Robert Barretto, Nick Juul, Tushar Desai, and Julien Sage for comments on the manuscript; and members of the Nusse lab for scientific and technical advice. This work was funded by the Howard Hughes Medical Institute (R.N.), the Virginia and D.K. Ludwig Fund for Cancer Research (R.N.), the Stanford Maternal and Child Health Research Institute (Ernest and Amelia Gallo Clinical Trainee Fellowship, P.V.W.), the Damon Runyon Cancer Research Foundation (Damon Runyon-Sohn Pediatric Cancer Fellowship Award, DRSG-28P-19, P.V.W.), the Burroughs Wellcome Fund (Career Award for Medical Scientists, Grant #1189428.01, P.V.W.), and the Stanford Department of Pathology (R.B.W.).

## Author contributions

P.V.W. and R.N. conceived the project, designed experiments, and wrote/edited the original manuscript. P.V.W. performed all experiments except the following: M.F. performed RNAscope in situ hybridization, F.K.H. assigned pathology to hepatoblastoma specimens, C.Z. prepared Smart-3SEQ libraries from micro-dissected samples, S.V. performed processing and alignment of Smart-3SEQ data, H.W. performed immunostaining of hepatoblastoma sections, and D.W. and J. C. performed 10x Visium library preparation, sequencing, and initial data processing. P.V.W. performed all other data curation, formal analyses, programming, and code implementation. J.P. provided instruction and resources, including sample code, for analysis of bulk RNA sequencing. M.M. and N.N. provided resources for scRNA sequencing. R.B.W. provided resources for and supervised the Smart-3SEQ experiments. R.N. supervised and provided funding for the work.

## Competing interests

R.N. is a board member of Bio-Techne and a member of the Scientific Advisory Board of Surrozen Inc. The authors declare no other competing interests.
