## [Peer Review File · Nature Communications]

REVIEWER COMMENTS

Reviewer #1 (Remarks to the Author): Clinical expert in paediatric hepatoblastoma, molecular biology, genomics, tumoroids, development, and single-cell RNA-seq

Article Summary:

In this manuscript by Wu et al., the authors argue that the Wnt target, Fgf19, is expressed by Sox4 cholangiocytic cells in hepatoblastoma tumors, and drives proliferation of surrounding tumor cells in a paracrine fashion. The authors perform laser capture microdissection and RNA sequencing of 17 patients with hepatoblastoma, resulting in a total of 72 samples that were of sufficient quality to analyze/interpret. They report that the embryonal histology had higher expression of Wnt targets (Axin2 and Dkk1), proliferation (MKi67), and cholangiocytic markers (Sox4). The focus of their manuscript is the relationship between these three functions of tumor cells.

Overall, this manuscript is of great interest to the field of hepatoblastoma and pediatric oncology researchers as the authors present intriguing data that dissects the mechanism by which tumor heterogeneity within HB tumors can result in functional growth advantages. There is incredible power with using tumor organoids to understand biology of the primary tumor.

Major points:

1. Fig 2: The overall finding that FGF19 is expressed within the embryonal components and not in the fetal components would benefit from strengthening the current data. The authors ascribe a significant amount of importance to FGF19 especially in the latter figures of the manuscript. The data that they provide showing higher amount of FGF19 in the embryonal component in Fig. 2a does indeed show an increase but there are embryonal samples that do not have higher Fgf19. Similarly, there is at least one sample in each of the fetal and normal livers that have moderate expression of Fgf19. What may help to convince me that this elevation of Fgf19 is a biologically relevant finding for tumor biology would be to show how frequent Fgf19-expressing clusters of tumor cells are present within a given tumor and across different patients. Many of the images show one region of a single tumor (e.g. Fig. 2c and e show the same region) and it is difficult to understand how truly representative these images are across different tumors.
2. Fig. 2: Though this reviewer appreciates the very clear in situ images, are there protein-level data that corroborate these findings?
3. Fig. 3: Do the tumor organoids only grow out fetal and embryonal cells or do they also grow out cells that have mesenchymal features? If yes, do the authors think that the tumor organoids are a fair representative of the original tumor?
4. Fig. 3: Would like to see confirmatory stains (in situ or IF) of the major features of the tumor organoids- FGFR4, Sox4, FGF19, etc.
5. Curious how the authors conceptualize the clusters of tumor organoids that express Fgf19. Is the proportion of FGF19-expressing cells in the tumor organoids higher/lower than what is seen in the original tumor.
6. How did the authors choose the level of FGF19 to add to the tumor cultures? Is it possible to determine whether the amount added to the organoids culture is anywhere physiologic/biologically relevant levels?
7. A major question is what is different about FGF19-expressing cells from those that do not express

FGF19 within tumor organoids. Since the authors have single cell RNA sequencing data, would it be possible to compare the DEGs between these clusters?

8. The authors use % colonies formed in the tumor organoid assays. Though this may represent proliferation as the title of the manuscript suggests, the differences in colony formation could also be due to enhanced survival, or lower cell death. There is correlative data with the proliferative marker Ki67 but I do not think that is sufficient for the authors to claim that "cholangiocytic differentiation drives cell proliferation..." A more accurate title would be that cholangiocyte differentiation enhances colony formation in HB tumor organoids... The authors could measure proliferation of the tumor organoids if they would like to bolster this claim. Proliferation data are included in Supp Figure 5 but it is unclear how this relates to percent colonies formed.

Minor points:

1. Why did the authors only choose to examine KLB in Figure 2 and not KLB and FGFR4?
2. Fig. 2: It would be useful to show the in situ images that are represented by Fig.2f
3. Out of curiosity, are the FGF19 clusters of cells malignant- do they carry the B-catenin mutation and express known HB tumor markers?

Reviewer #2 (Remarks to the Author): Expert in liver disease organoids, functional genomics, development, and single-cell RNA

This manuscript includes data describing the transcriptomic analyses on hepatoblastoma tumours. This paediatric cancer is induced by mutation affecting the WNT pathway. The authors perform laser dissection followed by RNA-Seq analyses. They show correlation between the tumour state and their transcription. In addition, they identify a number of genes specifically expressed in cancer cells including FGF19. They then derive organoids from hepatoblastoma and perform single cell analyses on the resulting lines grown in vitro. They identify a subgroup of line expressing high level of FGF19 and perform functional studies demonstrating that this growth factor can increase clonality of hepatoblastoma organoids. They then establish a link with Sox4 expression. This is an interesting study but several aspects need to be addressed before publication.

Figure 2b: it would be useful to show higher magnification of FGF19/Sox4 co-staining. Indeed, a large number of Sox4 positive cells seems to be located outside the FGF19 positive field.

Figure 3c: The authors should generate a table showing the number of cells per cluster for each patient.

The most interesting part of the manuscript is the derivation of Hepatoblastoma organoids. However, these organoids needs to be better characterised to reach standards used in the field. It would be incredibly useful to show immunostaining for basic markers i.e ALB/AFP/AHNF4/KRT19 but also for specific markers such as Sox4 and MKi67. The authors should also show specific foetal/embryonic markers identified in their single cell analyses. They should also provide some sort of growth curve for few lines over 5 passages. Finally, they should analyses key markers over prolonged period of time to

demonstrate phenotype stability (for example, ALB/AFP/HNF4 at passage 2, 5 and 10).

It is astonishing that the culture media does not contain A8301 a TGFb inhibitor which is commonly used to grow organoids. This is surprising especially since the Matrigel contains a lot of TGFb. This point is important for 2 reasons. First, TGFb blocks proliferation independently of WNT signalling and TGFb is an inducer of cholangiocytes differentiation. They need to explore this possibility.

The results describing the foetal/embryonic state of the organoids is very confusing. More analyses are needed:

- The comparison between in vitro grown hepatoblastoma organoids and primary tumour is difficult to follow. It seems that the organoids are a mix of both foetal/embryonic/mesenchymal cells while the tumours could be less heterogeneous. This is important to determine the impact of in vitro culture. In addition, this suggest these different states could be interchangeable in function of the environment.
- To define if the foetal / embryonic state corresponds to any developmental reality the authors should compare their hepatoblastoma organoids with hepatoblastoma organoid derived from foetal human liver (Wesley et al., Nature Cell Biology 2022). They could use the corresponding data to validate key markers.
- The approach used to define a foetal and embryonic signature is unclear. They need to provide the set of markers used to annotate the corresponding clusters. Again, the clusters currently identified are not different cell type but different state of the same cells.
- It would be incredibly useful to perform RNA velocity and pseudotime analyses to define if there is any relation between the different cell clusters.

“Since the hepatoblastoma tumoroids grew in serum-free media containing well-defined chemical components and recombinant growth factors”. This statement is misleading. The organoids are grown in Matrigel which contains a diversity of factors (even the reduced version) and thus their media is far from defined. Furthermore, they use media containing FBS to passage their cells. The authors need to correct this statement and take this aspect in consideration.

The data on the importance of FGF19 are simply not convincing:

- “HB4, HB15, and HB17 showed a high percentage of cells expressing FGFs, particularly FGF19, by single cell RNA sequencing” (page 17). This statement does not fit the data provide in Sup. Fig. 3d. HB6 or HB7 seems to contain similar level of cells expressing FGFs?
- Similarly, the knock down of FGF19 does affect the clonality but this remains extremely small difference and the stat analysis seems problematic (Supplementary Fig. S5b).
- The data provided only demonstrate the importance on clonality and not proliferation. Organoids don't survive single cell dissociation and thus the data only show that FGF19 might have a role in survival in suboptimal culture conditions. To demonstrate effect on proliferation, the authors need to knock down FGF19 after organoid formation and then count the cells 10 days later.
- MAPKinase pathway can be activated by a diversity of pathway especially insulin contains in the N2 used in their culture system. It would be super useful to provide Western blot analyses showing that addition of FGF19 does indeed increase the activity of this pathway.
- The molecular mechanisms seem to be only true for HB15 and thus can't be generalise.

- Sox4 knock down is extremely weak (and probably not statically significant see Fig. 6d). Furthermore, the single cell analyses show that most cells expressing FGF19 do not express Sox4. So, the impact on FGF19 expression could be due to technical challenges associated with the knock down. Sox4 knock down is also likely to induce cell death (and thus the effect on FGF19 is indirect).

Reviewer #3 (Remarks to the Author): Expert in hepatoblastoma genomics, functional genomics, organoids, development, and single-cell RNA-seq

This manuscript aims to better understand transcriptional heterogeneity in hepatoblastoma using histology-guided RNA sequencing in 17 tumors. This strategy showed different differentiation lineages as key determinants of intra-tumor heterogeneity. Further analysis of organoids identified FGF19 as a context-specific target of the WNT/ β -catenin activation related to increased cell proliferation.

1- Are the results obtained really novel? They should be carefully compared with recent transcriptomic HB classifications obtained by RNA sequencing and also with the previous C1/C2 classification. Results shown in Figure 1 match perfectly with these published transcriptomic classifications.

2- FGF19 over-expression is related to WNT/ β -catenin high activity and cholangiocytic differentiation: on which data is based (number of regions, statistical tests...)? Since it is a correlative result also in the tumoroid analysis, what the mechanism of FGF19 over-expression remains to be identified, sox4 depletion is insufficient to draw robust conclusion. Other transcription factors could be involved in the control of FGF19 promoter activity. Several other hypotheses should be tested.

3- Unsupervised analysis of single-cell data should be provided for each individual organoid and all together with an extensive identification of the tumor and non-tumor cell types. For comparison, RNA sequencing data and recent classifications (Nagae et al, Hirsch et al) should be used from pre and post-chemo samples. This is important since C1/C2 does not include the mesenchymal components.

4- Is FGF19 expression similar in the tumor sample compared to the derived corresponding tumoroid? What are the GI50 of both MEK inhibitors on HB cell lines?

RESPONSE TO REVIEWER COMMENTS

We thank the reviewers for thorough reading of our manuscript and thoughtful comments regarding the experimental and conceptual aspects of our work. Below we summarize the major additions that we have incorporated into our revision, followed by a point-by-point response to the reviewers' comments.

Several of the reviewers commented that our data on the pattern of FGF19 expression within tumor tissues was limited to a few images and requested that we provide further data to bolster our claims. We have now addressed this with examples of FGF19 RNA in situ hybridization in tumors from multiple different patients (Supplementary Fig. S3, S4) as well as spatial transcriptomics on two different hepatoblastomas and one normal liver (Fig. 2, Supplementary Fig. S2). The spatial transcriptomics data present convincing evidence not only of the validity of the fetal, embryonal, and mesenchymal gene signatures that we have identified by Smart-3SEQ but also of the colocalization of FGF19-expressing foci with AXIN2-high and SOX4-high embryonal regions of hepatoblastoma.

In addition, Reviewer 2 requested more characterization of our tumoroids. We have now included data on growth rates, bulk gene expression, as well as representative images of RNA in situ hybridization showing similar patterns of expression of FGF19, KLB, SOX4, and AXIN2 in our tumoroids (Fig. 3)

Reviewer 3 suggested additional comparisons of our single cell RNA sequencing data with previously published gene signatures associated with different classifications of hepatoblastoma, specifically as recently reported in Hirsch et al. 2021 and Nagae et al. 2021. We have performed analyses to determine how these various gene signatures overlay onto our scRNA seq data, now presented in Supplementary Fig. S6. We have additionally compared our scRNA seq data from hepatoblastoma tumoroids to the previously published scRNA seq data from primary hepatoblastomas in Song et al. 2022, showing correlation of subsets of our hepatoblastoma tumoroid cells with the tumor hepatoblast I and II signatures as well as the FGF19-expressing cells with the tumor neuroendocrine signature identified in this prior paper.

Several of the reviewers also noted that our functional data on the role of FGF19 in hepatoblastoma tumoroids was based on colony formation assays that may not reflect true proliferation. We have now included additional experiments in which we test the role of FGF19 on the growth and proliferation of intact tumoroids and show that indeed FGF19 stimulates proliferation of FGF19-negative tumoroids and that endogenous FGF19 is required for growth factor independent proliferation of FGF19-positive tumoroids (Fig. 5c, 6d).

We believe that these additional experiments have strengthened our conclusions overall. We have further addressed each of the reviewers' comments below.

Reviewer #1 (Remarks to the Author): Clinical expert in paediatric hepatoblastoma, molecular biology, genomics, tumoroids, development, and single-cell RNA-seq

Article Summary:

In this manuscript by Wu et al., the authors argue that the Wnt target, Fgf19, is expressed by Sox4 cholangiocytic cells in hepatoblastoma tumors, and drives proliferation of surrounding tumor cells in a paracrine fashion. The authors perform laser capture microdissection and RNA sequencing of 17 patients with hepatoblastoma, resulting in a total of 72 samples that were of sufficient quality to analyze/interpret. They report that the embryonal histology had higher expression of Wnt targets (Axin2 and Dkk1), proliferation (MKi67), and cholangiocytic markers (Sox4). The focus of their manuscript is the relationship between these three functions of tumor cells.

Overall, this manuscript is of great interest to the field of hepatoblastoma and pediatric oncology researchers as the authors present intriguing data that dissects the mechanism by which tumor heterogeneity within HB tumors can result in functional growth advantages. There is incredible power with using tumor organoids to understand biology of the primary tumor.

Major points:

1. Fig 2: The overall finding that FGF19 is expressed within the embryonal components and not in the fetal components would benefit from strengthening the current data. The authors ascribe a significant amount of importance to FGF19 especially in the latter figures of the manuscript. The data that they provide showing higher amount of FGF19 in the embryonal component in Fig. 2a does indeed show an increase but there are embryonal samples that do not have higher Fgf19. Similarly, there is at least one sample in each of the fetal and normal livers that have moderate expression of Fgf19. What may help to convince me that this elevation of Fgf19 is a biologically relevant finding for tumor biology would be to show how frequent Fgf19-expressing clusters of tumor cells are present within a given tumor and across different patients. Many of the images show one region of a single tumor (e.g. Fig. 2c and e show the same region) and it is difficult to understand how truly representative these images are across different tumors.

To address this and other reviewers' questions regarding the distribution of FGF19 across tumors and different patients, we have now provided multiple panels showing the pattern of FGF19 RNA in situ hybridization in tumors from 5 different patients. How frequently FGF19-expressing clusters occur in any given tumor depends on what percentage of the tumor is embryonal in histology, which in itself is challenging to quantify through morphologic assessment. Thus, we used 10x Visium to perform spatial transcriptomics on two different hepatoblastoma specimens with mixed histology to quantify the FGF19-expressing foci within a defined tumor region and, in a relatively unbiased manner, determine how closely this correlates with embryonal histology using the genes we identified by Smart-3SEQ to be differentially upregulated in the different histologic regions of hepatoblastoma. We now present convincing evidence in Fig. 2 and Supplementary Fig. S2 that FGF19 expression indeed colocalizes with multiple areas of embryonal histology within different tumors. Since FGF19-expressing spots comprise only ~2% of the tumor region assessed and ~8% of the embryonal regions, this explains the very low level of FGF19 expression detected by Smart-3SEQ and why

this did not reach statistical significance by the rigorous statistical methods for analyzing RNA sequencing data.

In regards to the question of whether the original images Fig 2c and 2e showed the same region, in fact, these images were from distinct patients (now labeled in Fig 2b-d as HB15, and Fig 2j-m as LCM-6).

2. Fig. 2: Though this reviewer appreciates the very clear in situ images, are there protein-level data that corroborate these findings?

We agree with the reviewer that protein-level expression data would be ideal and have provided immunofluorescence data where suitable antibodies are available. However, many of the antibodies available are not adequate for detecting their epitopes in FFPE sections. Specifically, FGF19, as a secreted protein, is challenging to detect in tissue sections and though we attempted IHC with available antibodies, we were not convinced of accurate representation of FGF19 protein expression as we could not reproducibly detect FGF19 by this method in normal human small intestine sections that serve as a positive control.

SOX4 and AXIN2 are also not reliably detected by antibodies in FFPE sections. Though we have access to a SOX4 antibody that detects SOX4 in methanol-fixed cells, with signal lost when SOX4 is knocked down by shRNA, this antibody does not work well for FFPE sections or immunoblot.

3. Fig. 3: Do the tumor organoids only grow out fetal and embryonal cells or do they also grow out cells that have mesenchymal features? If yes, do the authors think that the tumor organoids are a fair representative of the original tumor?

Our analyses of the tumoroid scRNA sequencing data indicates that they are primarily comprised of fetal and embryonal cells, with a population of cells that express both embryonal and mesenchymal genes, but none that express only mesenchymal-specific genes. Therefore, we have concluded that our tumoroids do not include a true mesenchymal component and represents only the epithelial components of the original tumor. Although these tumoroids can thus not be said to encapsulate all components of the original tumor, we do believe that these tumoroids are useful in investigating features of the epithelial components of the hepatoblastomas, while acknowledging that the mesenchymal components likely affect tumor cell behavior in ways that we cannot yet test.

4. Fig. 3: Would like to see confirmatory stains (in situ or IF) of the major features of the tumor organoids- FGFR4, Sox4, FGF19, etc.

We have now provided example images of RNA in situ detecting FGF19, KLB, SOX4, and AXIN2 in serial sections of FGF19-expressing tumoroids in Fig. 3.

5. Curious how the authors conceptualize the clusters of tumor organoids that express Fgf19. Is

the proportion of FGF19-expressing cells in the tumor organoids higher/lower than what is seen in the original tumor.

In early passage FGF19-expressing tumoroids, we visualize a low percentage of focal FGF19 across the tumoroids. However, as these tumoroids are passaged, we start to see tumoroids that are comprised entirely of FGF19-expressing cells or FGF19-negative cells that presumably arise from clonal proliferation (Fig. S5c). The proportion of FGF19-expressing cells in the tumor organoids, particularly after passaging, are not likely to faithfully represent what occurs in the original tumor since *in vitro* culture conditions inevitably differ from the *in vivo* environment. Nonetheless, we still think the FGF19-expressing tumoroids/cells are useful for answering specific biological questions addressed in this manuscript.

6. How did the authors choose the level of FGF19 to add to the tumor cultures? Is it possible to determine whether the amount added to the organoids culture is anywhere physiologic/biologically relevant levels?

We initially chose the amount of FGF19 to add (100ng/ml) to be the same as the amount of FGF10 normally added to the media. The amount of FGF10, in turn, was based on the standard concentration used in normal hepatocyte organoid cultures. Though we have not systematically determined the lowest concentration of FGF19 needed for colony formation/tumoroid growth, we have tested a lower amount of FGF19 (20ng/ml) and found no difference in colony formation rate at this 5-fold dilution. In an attempt to determine how much FGF19 is produced by FGF19-expressing cells, supposing that this is closer to what may be biologically relevant, we performed FGF19 ELISA on the media from FGF19-expressing tumoroids growing without exogenous growth factors and found a range between ~0.25-1.25 ng/ml produced at 24 hours after plating 100,000 total cells (of which only a fraction are FGF19-expressing). Given this approximation of the amount of FGF19 produced by the tumoroid cells, adding exogenous FGF19 at concentrations on the order of 10-100 ng/ml do not seem unreasonable. However, it is not possible to know what is a truly biologically relevant concentration of FGF19, since we also do not know the range of diffusion of FGF19 in tumor tissues.

7. A major question is what is different about FGF19-expressing cells from those that do not express FGF19 within tumor organoids. Since the authors have single cell RNA sequencing data, would it be possible to compare the DEGs between these clusters?

We have included the list of DEGs in the FGF19-expressing cell clusters in the Supplemental Data – in particular cluster 3, 9, 10 in Supp Data 4. Each of these clusters are marked by upregulation of AXIN2, SOX4, FGF19 and several other genes that we also identified as DEGs in the embryonal regions of hepatoblastoma by Smart-3SEQ, but there are additional genes that may be relevant to this unique subset of embryonal hepatoblastoma cells.

8. The authors use % colonies formed in the tumor organoid assays. Though this may represent proliferation as the title of the manuscript suggests, the differences in colony formation could

also be due to enhanced survival, or lower cell death. There is correlative data with the proliferative marker Ki67 but I do not think that is sufficient for the authors to claim that "cholangiocytic differentiation drives cell proliferation..." A more accurate title would be that cholangiocyte differentiation enhances colony formation in HB tumor organoids... The authors could measure proliferation of the tumor organoids if they would like to bolster this claim. Proliferation data are included in Supp Figure 5 but it is unclear how this relates to percent colonies formed.

We have now performed growth curves of intact tumoroids growing in the absence and presence of FGF19, to show that FGF19 indeed promotes tumoroid growth (Fig. 4, Supplementary Fig. S). The reason we have been performing colony formation assays in most cases is logistical due to the lower numbers of cells required for accurate colony formation assays (25,000 cells in 1 well for n=1 experiment) as compared to growth curves (50-100,000 cells/well x 3 wells = 150,000-300,000 cells to be able to count cells at day 3, 6, 9 for n=1 experiment).

Minor points:

1. Why did the authors only choose to examine KLB in Figure 2 and not KLB and FGFR4?

Published reports have indicated that, in normal tissues, KLB expression is more restricted than FGFR4 expression and is required for FGF19 signaling. Therefore we have focused on expression of KLB in our assays. Indeed, expression of FGFR4 in our tumoroids, as shown in Supplementary Figure 5S, indicated that FGFR4 is generally expressed in the same clusters of cells that express KLB.

2. Fig. 2: It would be useful to show the in situ images that are represented by Fig.2f

The quantitation of HNF4A/KRT19 staining in revised Fig. 2m is based on immunofluorescence as exemplified in Fig. 2l. Source images are available.

3. Out of curiosity, are the FGF19 clusters of cells malignant- do they carry the B-catenin mutation and express known HB tumor markers?

We have not yet been able to isolate the FGF19-expressing cells from either primary tumor sections or tumoroids, thus we have not formally proven that these are malignant tumor cells with the same b-catenin mutations at the rest of the tumor. However, all the analyses of RNA expression in these cells suggests that they express known hepatoblastoma markers and in particular of embryonal histology.

Reviewer #2 (Remarks to the Author): Expert in liver disease organoids, functional genomics, development, and single-cell RNA

This manuscript includes data describing the transcriptomic analyses on hepatoblastoma tumours. This paediatric cancer is induced by mutation affecting the WNT pathway. The authors

perform laser dissection followed by RNA-Seq analyses. They show correlation between the tumour state and their transcription. In addition, they identify a number of genes specifically expressed in cancer cells including FGF19. They then derive organoids from hepatoblastoma and perform single cell analyses on the resulting lines grown in vitro. They identify a subgroup of line expressing high level of FGF19 and perform functional studies demonstrating that this growth factor can increase clonality of hepatoblastoma organoids. They then establish a link with Sox4 expression. This is an interesting study but several aspects need to be addressed before publication.

Figure 2b: it would be useful to show higher magnification of FGF19/Sox4 co-staining. Indeed, a large number of Sox4 positive cells seems to be located outside the FGF19 positive field.

Higher magnification of the in situ results for FGF19 and SOX4 are now included in Fig. 2. As the reviewer points out, there are SOX4-positive cells outside of the FGF19-positive field, though it appears that FGF19 positive cells have slightly higher SOX4 expression identified both by in situ and scRNA sequencing of hepatoblastoma tumoroid cells.

Figure 3c: The authors should generate a table showing the number of cells per cluster for each patient.

This table has been included in revised Fig. 3.

The most interesting part of the manuscript is the derivation of Hepatoblastoma organoids. However, these organoids needs to be better characterised to reach standards used in the field. It would be incredibly useful to show immunostaining for basic markers i.e ALB/AFP/AHNF4/KRT19 but also for specific markers such as Sox4 and MKi67. The authors should also show specific foetal/embryonic markers identified in their single cell analyses. They should also provide some sort of growth curve for few lines over 5 passages. Finally, they should analyses key markers over prolonged period of time to demonstrate phenotype stability (for example, ALB/AFP/HNF4 at passage 2, 5 and 10).

In regards to this comment, we would like to emphasize that our tumoroids are derived from primary tumors from different patients, not from normal tissue, and therefore it is not unexpected that there is more variability in the tumoroids, just as tumors from different patient can vary greatly in biology from each other, as compared to normal tissues. Extensive work on 2D cancer cell lines has shown that tumor cells growing in culture often undergo selection over passages and cell lines grown in different labs can start to differ from the original cells. We thus expected that the tumoroids might be less phenotypically stable than organoids from normal tissues.

Nonetheless, we have now included data to address these points in revised Fig. 3. We have performed RNA in situ hybridization on tumoroids that express FGF19 to show co-localization with SOX4 and AXIN2, as well as inverse correlation with KLB. We have also performed qRT-PCR to show expression levels of known hepatoblastoma markers and hepatobiliary markers

as compared to normal liver. We have also provided growth curves for several tumoroids over several passages. We have also investigated expression of different hepatoblastoma markers and hepatobiliary markers (Supplementary Fig. S5b-c). The limitations of these experiments has been described in the text, noting that while the FGF19-negative tumoroids showed relatively stable expression of markers over time, the tumoroids with a subpopulation of FGF19-positive cells showed evidence of positive selection for FGF19-expressing cells over time. We therefore limited our experiments in this manuscript to early passage cells. Although we have performed some immunofluorescence staining of hepatobiliary markers, HNF4A, CK19, in the tumoroids, due to the heterogeneity within the tumoroids and between different tumoroids, we did not feel the patterns were representative and did not think they provided any more informative data than the single cell RNA sequencing.

It is astonishing that the culture media does not contain A8301 a TGFb inhibitor which is commonly used to grow organoids. This is surprising especially since the Matrigel contains a lot of TGFb. This point is important for 2 reasons. First, TGFb blocks proliferation independently of WNT signalling and TGFb is an inducer of cholangiocytes differentiation. They need to explore this possibility.

We appreciate the reviewer's attention to detail in noticing this mistake in our methods section in which we accidentally omitted the addition not only of the TGFb inhibitor A8301, but also mention of dexamethasone, and the growth factors, EGF, FGF10, and HGF, to the basal media used to grow our tumoroids. We have now corrected this section of the methods.

The results describing the foetal/embryonic state of the organoids is very confusing. More analyses are needed:

- The comparison between in vitro grown hepatoblastoma organoids and primary tumour is difficult to follow. It seems that the organoids are a mix of both foetal/embryonic/mesenchymal cells while the tumours could be less heterogeneous. This is important to determine the impact of in vitro culture. In addition, this suggest these different states could be interchangeable in function of the environment.

We have revised the description of our analyses in hopes that it will be less confusing to readers. We have found that the tumoroids are primarily a mix of fetal and embryonal hepatoblastoma cells, however it is difficult to determine just how representative the tumoroids are of their original tumor. Of note, we performed 10x Visium on the primary tumors from which the tumoroids HB4 and HB17 were derived. We were able to identify fetal, embryonal, and mesenchymal components in the primary tumors and the DEGs in each of these components did correlate with the DEGs associated with the different histologies as identified by Smart-3SEQ and also with DEGs in the clusters assigned as fetal or embryonal in the scRNA sequencing data from the tumoroids.

- To define if the foetal / embryonic state corresponds to any developmental reality the authors should compare their hepatoblastoma organoids with hepatoblast organoid derived from foetal

human liver (Wesley et al., Nature Cell Biology 2022). They could use the corresponding data to validate key markers.

We analyzed whether we could overlay the gene signatures associated with different hepatoblast differentiation states onto our scRNAseq data. We found that while most of the cells did express hepatoblast gene signatures (HB1 and HB2), the fetal hepatoblastoma cells correlated with the fetal hepatocyte gene signatures. We could not definitively identify the differentiation state of the embryonal FGF19-expressing hepatoblastoma cells. Though we noted that two of the fetal cholangiocyte DEGs, SOX4 and VIM, are overrepresented in the embryonal FGF19-expressing cells, on the other hand, of the cholangiocyte differentiation signatures, the hepatoblast 1 signature seemed to be most upregulated in this cluster of cells. The data suggest that these hepatoblastoma cells have diverged from the normal cholangiocyte differentiation trajectory, despite expression of the biliary transcription factor SOX4.

- The approach used to define a foetal and embryonic signature is unclear. They need to provide the set of markers used to annotate the corresponding clusters. Again, the clusters currently identified are not different cell type but different state of the same cells.

The gene lists used to define the fetal and embryonal signatures were obtained from the Smart-3SEQ laser capture microdissection based RNA sequencing. We have included lists of all the genes used to define the various signatures both Supplementary Table S2 and Supplementary Data 5.

- It would be incredibly useful to perform RNA velocity and pseudotime analyses to define if there is any relation between the different cell clusters.

We performed pseudotime analyses on our integrated scRNA sequencing dataset using Monocle, however the results depended on what cluster we hypothesize to be the starting point, shown in the Figure below. We omitted this analysis from the manuscript since we do not have experimental data to define which cluster of cells truly represents the cell of origin of the rest of the tumor.

Figure. Pseudotime analyses of tumoroid single cell data using Monocle, setting the starting point as either a) cluster with the highest Hepatoblast I signature (PCW 5) as defined by Wesley et al. 2022, b) embryonal specific hepatoblastoma cluster expressing FGF19, c) fetal specific hepatoblastoma cluster (most similar to normal liver), or d) C2 signature expressing cells.

“Since the hepatoblastoma tumoroids grew in serum-free media containing well-defined chemical components and recombinant growth factors”. This statement is misleading. The organoids are grown in Matrigel which contains a diversity of factors (even the reduced version) and thus their media is far from defined. Furthermore, they use media containing FBS to passage their cells. The authors need to correct this statement and take this aspect in consideration.

We have revised this statement to indicate that the media is relatively well-defined and is useful to determine the role of the recombinant growth factors that we normally add to the media. We do use 5% FBS to wash our tumoroid cells but remove nearly all the media containing FBS before resuspending the tumoroids in Matrigel and adding serum-free media, thus the final concentration of FBS to which the tumoroid cells are exposed is negligible. The fact that our FGF19-negative tumoroids do not grow in the basal media lacking exogenous recombinant growth factors suggests that any presence of these growth factors in the Matrigel or the low amount of FBS to which the tumoroids are exposed does not contribute to their growth.

The data on the importance of FGF19 are simply not convincing:

- “HB4, HB15, and HB17 showed a high percentage of cells expressing FGFs, particularly FGF19, by single cell RNA sequencing” (page 17). This statement does not fit the data provided in Sup. Fig. 3d. HB6 or HB7 seems to contain similar level of cells expressing FGFs?
- Similarly, the knock down of FGF19 does affect the clonality but this remains extremely small difference and the stat analysis seems problematic (Supplementary Fig. S5b).
- The data provided only demonstrate the importance on clonality and not proliferation. Organoids don't survive single cell dissociation and thus the data only show that FGF19 might have a role in survival in suboptimal culture conditions. To demonstrate effect on proliferation, the authors need to knock down FGF19 after organoid formation and then count the cells 10 days later.

We have revised the description of our data into separate analyses of FGF19-negative and FGF19-positive tumoroids and would humbly disagree with the reviewer in that we believe our cumulative data on primary tumors and tumoroids does add up to convincing evidence of the importance of FGF19 in promoting proliferation either when added exogenously to FGF19-negative cells or when expressed endogenously from FGF19-positive tumoroids. In general, we found a tight correlation between FGF19 expression, as shown in Supplementary Fig. S7 and whether the tumoroid cells required exogenous growth factors (Supplementary Fig. S8-S9). In addition, our data knocking down FGF19 in HB4, HB15, and HB17 show the importance of endogenous FGF19 in colony formation of these growth factor-independent tumoroids.

Regarding tumoroids HB6 and HB7, it was indeed also puzzling to us that these two tumoroids appeared to have a low level of FGF-expressing cells yet initially seemed to require exogenous growth factors for colony formation. Upon further testing of FGF19 expression, we found that these particular tumoroids were susceptible to variability of FGF19 expression after passaging

and when thawing from frozen stocks, thus later experiments would sometimes show growth factor-independence in colony formation as FGF19 expression also increased. We have excluded the data from these tumoroids due to the irreproducibility that is attributable to variable FGF19 expression at different passages/thawings. Overall, the data is still consistent with the claim that growth factor independence correlates with levels of FGF19 expressed endogenously by the tumoroids, but the level of FGF19 expression can vary with passaging/thawing.

Finally, we have now performed a growth curve that shows FGF19 knockdown slows growth of intact HB15 tumoroids in the absence of exogenous growth factors (Fig. 6d). Overall, we find the evidence demonstrating the requirement for FGF19 in FGF19-expressing cells to be convincing based on this new growth data in addition to the effect of FGF19 depletion on colony formation in three independent FGF19-expressing tumoroids (HB15, HB4, HB17), with statistical analyses of three independent experiments.

- MAPKinase pathway can be activated by a diversity of pathway especially insulin contains in the N2 used in their culture system. It would be sper useful to provide Western blot analyses showing that addition of FGF19 does indeed increase the activity of this pathway.

Indeed, MAPK activation can be demonstrated by phospho-ERK Western blot analyses. Although we see robust phospho-ERK activity in tumoroids growth in media containing all the growth factors or FGF19 alone, because the tumoroids do not grow in the absence of growth factors, it has been challenging to obtain enough cells in the absence of growth factors to show that these have decreased phospho-ERK. We have instead performed the converse experiment in which we have inhibited FGF19 signaling using a FGFR4 inhibitor, BLU9931, and show that phospho-ERK is reduced (Fig. S9F).

We have not ruled out that other factors such as insulin do also contribute to basal MAPK activation to some extent. In our experiments withdrawing growth factors, we often need to add a MEK inhibitor to fully arrest cells in G0/G1 over a period of 36 hours suggesting that perhaps there is some basal MAPK activation. However, the argument against a sustained role for other factors in MAPK activation is in our data showing that tumoroids plated in the absence of EGF/FGF/HGF ultimately are unable to increase their cell number over days in culture and eventually undergo cell death.

- The molecular mechanisms seem to be only true for HB15 and thus can 't be generalise.

We have shown FGF19 expression occurs in nearly ~50% of all primary hepatoblastomas and reproducibly in at least 3 of the tumoroids we derived – HB4, HB15, HB17. We have shown that FGF19 is required in all three of these tumoroids both by shRNA depletion and by inhibition with an FGFR4 inhibitor. In terms of the mechanism by which FGF19 is induced, we have now shown in HB17 as well, that FGF19 expression also depends on Wnt/beta-catenin and SOX4 (Fig. S10h-j). Thus, we do believe that our findings can be generalizable to FGF19-

expressing hepatoblastoma cells; we are only limited by the rare sample size and the variability in lentiviral transduction efficiency of different tumoroids.

- Sox4 knock down is extremely weak (and probably not statically significant see Fig. 6d). Furthermore, the single cell analyses show that most cells expressing FGF19 do not express Sox4. So, the impact on FGF19 expression could be due to technical challenges associated with the knock down. Sox4 knock down is also likely to induce cell death (and thus the effect on FGF19 is indirect).

We have now repeated the SOX4 knockdown multiple times with significance demonstrated for the effect on FGF19 expression (Fig. 7d). We have also shown that while it is challenging to overexpress SOX4 to a high level, we can demonstrate statistically significant increases in expression of the known target gene TEAD2 as well as FGF19 (Fig. 7g). We cannot rule out the alternative model that SOX4 is a survival factor for FGF19-expressing cells and point out this possibility in the discussion. However, even if the effect is indirect, SOX4 is nonetheless required for the FGF19-expressing population to exert its pro-proliferative effect on the FGF19-negative tumor cells.

Reviewer #3 (Remarks to the Author): Expert in hepatoblastoma genomics, functional genomics, organoids, development, and single-cell RNA-seq

This manuscript aims to better understand transcriptional heterogeneity in hepatoblastoma using histology-guided RNA sequencing in 17 tumors. This strategy showed different differentiation lineages as key determinants of intra-tumor heterogeneity. Further analysis of organoids identified FGF19 as a context-specific target of the WNT/ β -catenin activation related to increased cell proliferation.

1- Are the results obtained really novel? They should be carefully compared with recent transcriptomic HB classifications obtained by RNA sequencing and also with the previous C1/C2 classification. Results shown in Figure 1 match perfectly with these published transcriptomic classifications.

While the gene signatures that we obtain in Figure 1 do overlap with several published transcriptomic classifications, we would argue that the rest of the findings in our manuscript are indeed novel in linking FGF19 and the biliary transcription factor SOX4 to the heterogeneous outcomes in proliferation and gene expression in response to Wnt activation. We have compared our single cell RNA sequencing data from hepatoblastoma tumoroids to many of the published classifications (Supplementary Fig. S6), and there are notable similarities and differences between these and our fetal/embryonal/mesenchymal classifications obtained from laser capture microdissection of pathologist-confirmed histologic components of hepatoblastoma. We do believe that it is useful to clarify these differences and identify which genes are truly reflective of the different histologic phenotypes.

2- FGF19 over-expression is related to WNT/ β -catenin high activity and cholangiocytic

differentiation: on which data is based (number of regions, statistical tests...)? Since it is a correlative result also in the tumoroid analysis, what the mechanism of FGF19 over-expression remains to be identified, sox4 depletion is insufficient to draw robust conclusion. Other transcription factors could be involved in the control of FGF19 promoter activity. Several other hypotheses should be tested.

We have now used multiple different approaches – Smart-3SEQ, RNA in situ, spatial transcriptomics via 10x Visium – that have confirmed the correlation between FGF19 expression with high Wnt/b-catenin and high SOX4. There is inherent variability in the number of FGF19-expressing foci in tissues depending on the relative amount of embryonal tumor tissue captured, so it is inherently difficult to compare FGF19 expression in a meaningful way across different tumors. The Smart-3SEQ technique was unable to demonstrate statistical significance in FGF19 upregulation in the embryonal regions due to the very low expression levels and the variability in what portion of an embryonal region was dissected by laser capture microdissection. However, the high AXIN2 and SOX4 expression in the embryonal regions were statistically significant. Although we only were able to perform Visium successfully on 2 hepatoblastoma sections, we were able to assign the embryonal components based on AXIN2 and SOX4 expression as well as expression of other embryonal genes identified by Smart-3SEQ. Using a chi-squared test for FGF19+/- vs. embryonal/non-embryonal as quantified by dots on the Visium slide, we ultimately did find a statistically significant relationship between FGF19 expression and embryonal/AXIN2-high/SOX4-high cells.

Regarding the concern that SOX4 depletion is insufficient to draw a robust conclusion about the mechanism of FGF19 expression, it is true that we have not ruled out alternative models, which we have now elaborated on in the revised discussion. However, we have now repeated the SOX4 knockdown multiple times with significance demonstrated for the effect on FGF19 expression. We have also shown that while it is challenging to overexpress SOX4 to a high level, we can demonstrate statistically significant increases in expression of the known target gene TEAD2 as well as FGF19. Thus, as we write in the discussion:

“We favor a model whereby SOX4 and b-catenin cooperate to induce expression of FGF19 through binding at its promoter. This is based on a significant body of prior work suggesting that SOX transcription factors interact with b-catenin in transcriptional complexes to modulate the outcome of Wnt activation. Nonetheless, our experiments have not formally excluded the alternative model that SOX4 and b-catenin indirectly promote expression of FGF19 through other downstream transcription factors or that FGF19-expressing cells uniquely require SOX4 and b-catenin for survival, explaining why depleting these genes reduces expression of *FGF19*.”

3- Unsupervised analysis of single-cell data should be provided for each individual organoid and all together with an extensive identification of the tumor and non-tumor cell types. For comparison, RNA sequencing data and recent classifications (Nagae et al, Hirsch et al) should be used from pre and post-chemo samples. This is important since C1/C2 does not include the mesenchymal components.

We have now included a supplemental figure (Fig S7) with separate UMAP plots for the scRNA sequencing data for each individual tumoroid, with visualization of the expression of the major genes of interest discussed in this paper. As this data appeared to recapitulate the integrated analyses that we presented in Figure 4, we do not believe that additional exhaustive analyses of each of the tumoroids separately would add significantly to the current manuscript. Overall, our analyses indicate the presence of only tumor cells in our tumoroids and not a significant or detectable number of non-tumor normal cell types. We have uploaded all the raw data to the GEO public repository and hope that making this data available to other researchers will encourage future analyses that may lead to new discoveries beyond what we have presented in our current manuscript.

As suggested by this reviewer, we have further used the gene lists corresponding to the hepatocytic, proliferative, and mesenchymal hepatoblastomas in Nagae et al., as well as the hepatocytic, progenitor, and mesenchymal hepatoblastomas in Hirsch et al to calculate the signature score for each of the cells in our hepatoblastoma tumoroids (Supplementary Fig. S6). Overall, the expression of these gene signatures was weak across our tumoroids. In fact, individual marker genes from any of these gene lists were often expressed in different populations of cells within our hepatoblastoma tumoroids, which is not surprising given that the prior gene lists were obtained from bulk RNA sequencing of tumors and thus likely comprised mixtures of the cell types found in our tumoroids. Nonetheless, we found that the hepatocytic signatures described in these prior publications were most highly expressed in what we defined as the fetal hepatoblastoma cells. The mesenchymal signature from Hirsch et al. corresponded best with our FGF19-expressing embryonal cells, while the mesenchymal signature from Nagae et al. was not significantly detected. We have incorporated this data into Fig S6 and discussed the analyses in the main text. However, we overall do not believe that these gene signatures are best used for the purposes of classifying single cell RNA sequencing data, due to the caveats above, and rather the gene lists we obtained from our own Smart-3SEQ analyses performed better in classifying the cell populations within our tumoroids as fetal, embryonal, or mesenchymal.

Finally, we also compared our single cell RNA sequencing data with that of Song et al. 2022. Here we do see good correlation between our fetal hepatoblastoma cells and what they report as hepatoblast II (derived from a pure fetal hepatoblastoma). We also see interestingly a correlation between the FGF19-expressing embryonal cells and what they define as a neuroendocrine tumor cell population.

4- Is FGF19 expression similar in the tumor sample compared to the derived corresponding tumoroid? What are the GI50 of both MEK inhibitors on HB cell lines?

We have performed Visium spatial transcriptomics on original tumor sections from which we derived the FGF19-expressing tumoroids HB4 and HB17, showing FGF19 expression that correlates with high Axin2 and high Sox4 expression, as we see in the tumoroids (Fig 2, Supplementary Fig S2). It is not possible to truly correlate the FGF19 expression level in

tumoroids as compared to original tumor, since this depends on what part of the tumor is sampled for the tumoroid, eg. whether this is an FGF19-expressing part or not.

We tested the effect of both MEK inhibitors, U0126 and trametinib, on both colony formation and tumoroid growth and find good correlation between the two assays, with GI50 of ~6 μ M for U0126 and ~16nM for trametinib (Supplementary Fig. S9).

Reviewers' comments:

Reviewer #1 (Remarks to the Author):

We thank the authors for addressing many of my concerns. The manuscript has improved considerably, and I appreciate the additional analysis performed. Despite these improvements, I have two main concerns about the manuscript:

1. The title is overstated in its current form. The authors have not shown that Sox4 drives proliferation through a Wnt-dependent FGF-19 signaling pathway. I would expect to see changes in Wnt signaling/activation when SOX4 is inhibited. Instead, we see no changes to CTNNB1 as a result of Sox4 knockdown. Also, these findings are specific to embryonal HB. A more appropriate title would be, Sox4 may cooperate with Wnt-dependent FGF19 signaling to promote cell proliferation in embryonic HB.
2. There are claims about associations with FGF19+ clusters that are not convincing with the data provided. I would be convinced of the importance of these associations if the authors demonstrate the frequency by which these associations occur and include more than one image from one patient to demonstrate that these associations are unique to FGF19+ clusters.

MAJOR POINTS:

Figure 2c,d: It is very difficult to know whether the claims made by the authors regarding colocalization is seen in the majority vs minority of FGF19+ clusters. Do all FGF19 clusters colocalize with AXIN2 and SOX4? If so, this should be explicitly stated. If not, a ratio should be reported. Since these associations serve as the foundation of the studies performed in tumor organoids, it is very important that the authors show the frequency re co-localization and associations. As it currently stands, I presume only one region from one patient is shown for AXIN2 and SOX4 in Fig. 2d.

Fig. 2j: Which pattern of B-catenin stain are the authors showing in Fig. 2j (third image from the left)? This looks like high nuclear B-catenin staining. In which figure panel are the authors showing that Nuc>Memb staining surrounds the FGF19 cluster (line 231). I think that the authors are highlighting the nuc>mem B-catenin in the rightmost image of Fig. 2j. These cells in green are surrounding the FGF19 cluster but they appear evenly distributed in the image provided (i.e. are they concentrated around the FGF19 cluster?). This is confusing. It seems like the authors are making an important claim that High nuc (currently shown as co-localized with FGF19+) is not associated with Ki67 and Nuc>Mem is associated with Ki67. Unlike what the authors claim, I do not think the Ki67 + cells (purple cells) in Fig.2j are the same cells as the nuc>mem B-catenin cells (fluorescent green). Again, these associations seem to be the bedrock of this paper yet I am left confused with the associations the authors are claiming and the images provided.

Fig. 6e: This is a really interesting result. That CM from FGF19-expressing tumoroids can promote growth of FGF19-negative tumoroid cells. In order to implicate FGF19 as the soluble factor that is responsible for this growth, the authors need to show that CM from FGF19-shRNA does not result in enhanced tumoroid growth of FGF19-negative cells.

Line 209: Did the regions identified by Smart-3SEQ as fetal, embryonal, and mesenchymal correlate with

the histology/morphology of these regions. I presume so but this should be explicit here because the histology/morphology to characterize these regions remains the gold standard.

Line 229: What does this mean to have "particularly high" nuclear staining? This seems quite subjective and imprecise. How is that different than nuclear B-catenin staining? This is confusing as to what significance the authors are ascribing to this finding.

Line 438: I believe the more accurate interpretation is that "These phenotypes were partially rescued..."

MINOR POINTS:

Page 8. Last sentence of paragraph should read "... we subsequently focused on understanding the relationship between these features in embryonal HB."

Line 203: It is a stretch to call Axin2 and Sox4 as "markers" of embryonal histology. Rather they are upregulated in the embryonal regions as they are still expressed in the other regions.

Legend for Fig. 2: Panel m is misslabelled as f

Reviewer #2 (Remarks to the Author):

The authors have provided limited answer to the original comments. Some aspects are still problematic. The comments did raise a number of concerns which are confirmed by the new data provided. However, the authors have not considered these limitations in their data interpretation or their conclusions.

1. Figure 2b: it would be useful to show higher magnification of FGF19/Sox4 co-staining. Indeed, a large number of Sox4 positive cells seems to be located outside the FGF19 positive field.

Higher magnification image confirms that only a minority of cells co-expressing Sox4/FGF19. The authors need to change their conclusion accordingly.

2. Figure 3c: The authors should generate a table showing the number of cells per cluster for each patient.

The table is provided figure 4?

3. The most interesting part of the manuscript is the derivation of Hepatoblastoma organoids. However, these organoids need to be better characterised to reach standards used in the field. It would be incredibly useful to show immunostaining for basic markers i.e ALB/AFP/AHNF4/KRT19 but also for specific markers such as Sox4 and MKi67. The authors should also show specific foetal/embryonic

markers identified in their single cell analyses. They should also provide some sort of growth curve for few lines over 5 passages. Finally, they should analysed key markers over prolonged period of time to demonstrate phenotype stability (for example, ALB/AFP/HNF4 at passage 2, 5 and 10).

The authors did not provide immunostaining but in situ analyses. This is useful but do not fit the usual standard of characterisation. IF analyses on Alb/AFP/CK19 are very common in the field of liver organoids. They have to be able to provide such staining. Also, their comments on the lack of stability of their organoid need to be considered in their interpretation of their data and discussed in their conclusion. Again, the original comment did raise some concerns which is then confirmed by the authors but who chose to ignore it. Importantly, it seems that their culture selection ultimately selects for Cholangiocytes/biliary cells and they could lose the original hepatoblastoma. This could strongly affect their interpretation regarding the importance of FGF19. They need to show co-staining of CK19/AFP or CK19/ALB over a time course to determine the fraction of biliary/hepatoblastoma cells.

4. To define if the foetal / embryonic state corresponds to any developmental reality the authors should compare their hepatoblastoma organoids with hepatoblast organoid derived from foetal human liver (Wesley at al., Nature Cell Biology 2022). They could use the corresponding data to validate key markers.

They authors did the comparison but show very little data. It seems that the profile does not overlap suggesting that their interpretation of foetal/embryonic signature is not accurate. Why not develop this aspect further? Again, the signature suggest a very strong biliary contamination which is likely to rapidly over grow the tumour cells.

5. The approach used to define a foetal and embryonic signature is unclear. They need to provide the set of markers used to annotate the corresponding clusters. Again, the clusters currently idenfied are not different cell type but different state of the same cells.

See comment 4. They need to use previous publications on foetal tissue to annotate their clusters.

6. It would be incredibly useful to perform RNA velocity and pseudotime analyses to define if there is any relation between the different cell clusters.

Again, they did not provide RNA velocity analyses but did perform pseudotime analyses. Pseudotime always requires a start point. However, the trajectory should be informative. If not, their interpretation of foetal/embryonic is not accurate. This could be a mix of cells without any link.

7. "Since the hepatoblastoma tumoroids grew in serum-free media containing well-defined chemical components and recombinant growth factors". This statement is misleading. The organoids are grown in Matrigel which contains a diversity of factors (even the reduced version) and thus their media is far from defined. Furthermore, they use media containing FBS to passage their cells. The authors need to correct this statement and take this aspect in consideration.

Again, the authors should do an effort on this aspect. This is very simple. Matrigel contains growth factors which can influence their results and FBS is a problem. Why use FBS on the first instance?

Organoids usually do not require FBS for passaging etc.

8. The data on the importance of FGF19 are simply not convincing:

- "HB4, HB15, and HB17 showed a high percentage of cells expressing FGFs, particularly FGF19, by single cell RNA sequencing" (page 17). This statement does not fit the data provide in Sup. Fig. 3d. HB6 or HB7 seems to contain similar level of cells expressing FGFs?
- Similarly, the knock down of FGF19 does affect the clonality but this remains extremely small difference and the stat analysis seems problematic (Supplementary Fig. S5b).
- The data provided only demonstrate the importance on clonality and not proliferation. Organoids don't survive single cell dissociation and thus the data only show that FGF19 might have a role in survival in suboptimal culture conditions. To demonstrate effect on proliferation, the authors need to knock down FGF19 aler organoid formation and then count the cells 10 days later.

These comments are really essentials. The authors answer them by excluding some of their original data to make them more consistent. Their justification is acceptable but again should impact the interpretation of their results. The growth cure is key. However, this experiment includes only 1 cell line with not statistical analyses (see figure 6d)?

9. MAPKinase pathway can be activated by a diversity of pathway especially insulin contains in the N2 used in their culture system. It would be super useful to provide Western blot analyses showing that addition of FGF19 does indeed increase the activity of this pathway.

The requested experiments were not performed. The justification on cell number is difficult to follow. You can grow organoid in optimal condition to obtain enough cells and then starve them from specific growth factors for 1-2 hour (ERK phosphorylation change very rapidly) and then re-add specific combination of factors. In any case, their experiment show that other factors activate ERK and the authors should discuss this aspect in their conclusion.

10. The molecular mechanisms seem to be only true for HB15 and thus can 't be generalise.

The author's answer does not fit the data provided. Again, a large number of functional analyses are only performed on HB15 based on their figure legend. Their results are are sometimes very variable and are not reproducible across organoid lines which have nonetheless being carefully selected to fit their hypothesis. This could indeed be due to technical challenges and tumour variabilities. However, the authors must recognise these problems and tune down the generalisation of their conclusions.

11. Sox4 knock down is extremely weak (and probably not staGcally significant see Fig. 6d). Furthermore, the single cell analyses show that most cells expressing FGF19 do not express Sox4. So, the impact on FGF19 expression could be due to technical challenges associated with the knock down. Sox4 knock down is also likely to induce cell death (and thus the effect on FGF19 is indirect).

The knock down of Sox4 is still limited to 50% but stat analyses are now provided. However, the exact stat test used is not provided in the figure legend (Figure 7.d). How many experiments were performed? Which cell line? Which T-test? Can they really use paired comparison if those are independent experiments? , there is no error bar on the control? etc. They could have analyses cell death level after

knock down and show decrease in CK19 expressing cells etc. Nonetheless, the authors did include this possibility in their revised conclusion.

Reviewer #3 (Remarks to the Author):

The authors partly to my comments, modifications are challenging to identify since there is tracking and figures not numbered.

1-Comparison with transcriptomic classifications previously published:

Utilizing recent RNAseq classification as a reference is essential and more precise than C1/C2. Inferring classification obtained from bulk transcriptomic in single-cell analysis is important, and most studies are performed in that direction. Organoids add complexity, but using classifications obtained in primary tumors as a reference is essential.

- the present single-cell results match more with the Hirsch et al. classification; it should be included in the principal figure in addition, or replacing, C1/C2.
- The lack of similarity with Nagae's classification is surprising it could be due to the choice of genes that are not well explored in single-cell data.

-

2-Functional analysis of FGF19 over-expression is still limited.

RESPONSE TO THE REVIEWERS

We thank the reviewers for rereading our manuscript and providing additional thoughtful comments. We noted that several of the reviewers' comments appeared to neglect additional data that we had included in the prior round of revision, particularly data we had included in Supplementary Information. **For clarity, we have moved much of this supplementary data to the main figures, along with additional experiments addressing all the reviewers' comments from the 1st and 2nd round of reviews.**

To highlight a few points:

- In addressing **reviewer 1's** main concerns about reproducibility of FGF expression across different patients, we had included in the Supplementary Info multiple additional images of FGF expression in different patients and additional **new Visium spatial transcriptomics experiments** showing FGF expression in other samples. It seemed that these new data were missed by the reviewer, and have now been moved to the main Fig. 1-2. **We have also added additional quantitation and images from other patients in Fig. 2 and S3 that corroborate our findings.**
- We found that many of **reviewer 2's** comments suggested that they did not take into consideration much of our additional data provided in Supplementary Info. First, in response to the reviewer's request, **we had performed analyses of our single cell RNA sequencing data using markers of cell types in the embryonic liver, but this went unnoticed by the reviewer.** We have now moved these data (previously in Supplemental Info) to Fig. 4g. **We found no clear expression of the adult hepatocyte and cholangiocyte signatures, suggesting that our tumoroids were not contaminated with either normal hepatocytes or cholangiocytes.**
- In addition, in response to **reviewer 2**, **we have now added immunofluorescence imaging of our tumoroids (Fig. 3h-i)**, which confirms the results from the scRNA sequencing and qPCR analyses that we had performed, showing that hepatic and cholangiocyte markers are maintained across passaging in the majority of tumoroids. In the small subset of tumoroids that become dominated by cholangiocyte tumor cells over passaging, we note that these remain indeed tumor cells with biliary differentiation, not contamination by normal bile duct organoids (as clarified in Fig. 4g), which seemed to be a misunderstanding of the reviewer. We also note that we performed all the functional experiments on the tumoroids at low passage and therefore believe our conclusions are sound, however **we now include a section on limitations to our study in the conclusions.**
- Finally, regarding **reviewer 2's** comment that "a large number of functional analyses are only performed on HB15 based on their figure legend", **we had previously included functional analyses on HB4 and HB17 in supplementary figures, which we have now moved to the main Figs. 6 and 7.**
- **Reviewer 3** also appeared not to have noticed our supplemental data which showed our single cell RNA sequencing data overlaid with the Hirsch et al. and Nagae et al. gene signatures. **We have now moved this data, along with additional analyses, into main Fig. 4f**, and believe that the revised figure is a clearer representation of how our data compares with previously published signatures.

We believe that these modifications and additional experiments have strengthened our conclusions overall. We have further addressed each of the reviewers' comments point-by-point below.

Reviewers' comments:

Reviewer #1 (Remarks to the Author):

We thank the authors for addressing many of my concerns. The manuscript has improved considerably, and I appreciate the additional analysis performed. Despite these improvements, I have two main concerns about the manuscript:

1. The title is overstated in its current form. The authors have not shown that Sox4 drives proliferation through a Wnt-dependent FGF-19 signaling pathway. I would expect to see changes in Wnt signaling/activation when SOX4 is inhibited. Instead, we see no changes to CTNNB1 as a result of Sox4 knockdown. Also, these findings are specific to embryonal HB. A more appropriate title would be, Sox4 may cooperate with Wnt-dependent FGF19 signaling to promote cell proliferation in embryonic HB.

We have adjusted our title to incorporate this reviewer's suggestions. To elaborate, our title was not meant to imply a linear pathway whereby SOX4 drives Wnt activation, that then promotes FGF19 expression. Rather, we show that expression of FGF19 in a subset of cells is downstream of and dependent on both SOX4 and Wnt/ β -catenin.

2. There are claims about associations with FGF19+ clusters that are not convincing with the data provided. I would be convinced of the importance of these associations if the authors demonstrate the frequency by which these associations occur and include more than one image from one patient to demonstrate that these associations are unique to FGF19+ clusters.

In response to this major concern expressed by the reviewer, we need to point out that we had added several results in the prior revision of the manuscript that the reviewer may have overlooked:

- 1) We had executed new spatial transcriptomics using 10x Visium of two different patients' tumors (HB4 and HB17), strengthening the association of FGF19 expression with a subset of cells within the AXIN2^{high} and SOX4^{high} embryonal components of the tumors (now shown in Fig. 1g-h, 2b-c, Supplementary Fig S2a-e).**

Fig 2b-c:

Supplementary Fig S2a-e:

2) In the prior submission, we had included several images showing FGF19 expression in tumors from other patients in addition to HB4, HB15, and HB17 (now Supplementary Fig. S4 and S5). We had also included in situ hybridization experiments showing colocalization with AXIN2 and SOX4 in HB15 (Fig. 2f) and have now added images from HB4 and HB17 (Fig. 2e, Supplementary Fig. 2g).

- 3) In addition to the data included in the previous submission, we have now added quantification of the colocalization between FGF19 and AXIN2 and SOX4 expressing cells (Fig. 2g-h).

MAJOR POINTS:

Figure 2c,d: It is very difficult to know whether the claims made by the authors regarding colocalization is seen in the majority vs minority of FGF19+ clusters. Do all FGF19 clusters colocalize with AXIN2 and SOX4? If so, this should be explicitly stated. If not, a ratio should be reported. Since these associations serve as the foundation of the studies performed in tumor organoids, it is very important that the authors show the frequency re co-localization and associations. As it currently stands, I presume only one region from one patient is shown for AXIN2 and SOX4 in Fig. 2d.

As we mentioned above, we have included images from multiple patients showing colocalization of FGF19+ cells with AXIN2 and SOX4 in serial sections or via double in-situ (Fig. 2e, Fig. 2f, and Supplementary Fig. 2g, shown above). We have also quantified the colocalization of FGF19 with

AXIN2 and SOX4 using double in situ, showing that indeed, FGF19 expression is found almost exclusively in AXIN2+ or SOX4+ cells (Fig. 2g-h, shown above). Finally, we have also included, as we did in our prior revision, the quantification of FGF19 co-localization with AXIN2^{high} and SOX4^{high} regions by a 2nd method, 10x Visium (now shown in Fig. 1g-h, 2b-c, Supplementary Fig S2a-e, shown above).

Fig. 2j: Which pattern of B-catenin stain are the authors showing in Fig. 2j (third image from the left)? This looks like high nuclear B-catenin staining. In which figure panel are the authors showing that Nuc>Memb staining surrounds the FGF19 cluster (line 231). I think that the authors are highlighting the nuc>mem B-catenin in the rightmost image of Fig. 2j. These cells in green are surrounding the FGF19 cluster but they appear evenly distributed in the image provided (i.e. are they concentrated around the FGF19 cluster?). This is confusing. It seems like the authors are making an important claim that High nuc (currently shown as co-localized with FGF19+) is not associated with Ki67 and Nuc>Mem is associated with Ki67. Unlike what the authors claim, I do not think the Ki67 + cells (purple cells) in Fig.2j are the same cells as the nuc>mem B-catenin cells (fluorescent green). Again, these associations seem to be the bedrock of this paper yet I am left confused with the associations the authors are claiming and the images provided.

To address this reviewer’s comments, we have first changed the nomenclature to be clearer, renaming the three categories of cells according to the level of nuclear β -catenin – low, med, or high. We now show high magnification images of the β -catenin and DAPI staining in these nuclear β -catenin low (1), med (2), and high (3) cells, and indicate using numbered boxes which cells we are referring to in the images (Fig. 2i).

We note that the intensity of β -catenin staining within the nuclear regions in box 2 (Med) is higher than that in box 1 (Low) and lower than that in box 3 (High). In addition, the Low nuclear β -catenin staining cells (1) are distinctive in that they exhibit significant membranous β -catenin staining. The High nuclear β -catenin staining cells (3) colocalize with the outlined FGF19-expressing cells, while the Med cells (2) are in the layer of cells adjacent to the FGF19-expressing cluster. As the reviewer points out, these Ki67 expressing cells are not right next to the FGF19-expressing cells, suggesting that FGF19 can diffuse through several cell thicknesses and is not necessarily acting only on the immediately surrounding cells.

Fig. 2i:

Fig. 6e: This is a really interesting result. That CM from FGF19-expressing tumoroids can promote growth of FGF19-negative tumoroid cells. In order to implicate FGF19 as the soluble factor that is responsible for this growth, the authors need to show that CM from FGF19-shRNA does not result in enhanced tumoroid growth of FGF19-negative cells.

This is a good suggestion, but unfortunately the suggested experiment cannot be performed – it is impossible to obtain CM media from FGF19-KD cells that does not contain other added growth factors, as these cells do not survive in the absence of growth factors. Instead, the evidence suggesting that FGF19 is indeed one of the main soluble factors required for proliferation of FGF19-negative cells comes from the subsequent experiment where we knock down FGF19 in FGF19-expressing tumoroids, which contain both FGF19-positive and FGF19-negative cells. We show that colony formation/proliferation in the absence of exogenous growth factors is reduced when FGF19 is knocked down in FGF19-expressing tumoroids.

Line 209: Did the regions identified by Smart-3SEQ as fetal, embryonal, and mesenchymal correlate with the histology/morphology of these regions. I presume so but this should be explicit here because the histology/morphology to characterize these regions remains the gold standard.

Yes, the regions selected for laser capture microdissection and Smart-3SEQ were identified based on histology/morphology by a board-certified pediatric pathologist. We have made this point more explicit in the text. Furthermore, the markers identified by Smart-3SEQ for fetal/embryonal/mesenchymal components were then independently validated by 10x Visium spatial transcriptomics on two independent specimens where the histologies were confirmed by morphology.

Line 229: What does this mean to have "particularly high" nuclear staining? This seems quite subjective and imprecise. How is that different than nuclear B-catenin staining? This is confusing as to what significance the authors are ascribing to this finding.

As in comments above, we now include high magnification images of the β -catenin and DAPI staining in the nuclear β -catenin low, med, and high cells, and indicate using numbered boxes which cells we are referring to in the images (Fig. 2i, shown above).

We acknowledge that making these assignments by eye can be subjective. To address this reviewer's concerns, we alternatively used an automated method to quantitate nuclear β -cat and Ki67 and show that we obtain similar results (Supplementary Fig S3a-d).

Supplementary Fig. S3a-d:

Line 438: I believe the more accurate interpretation is that "These phenotypes were partially rescued..."

We have revised this sentence to reflect this reviewer's comment.

MINOR POINTS:

Page 8. Last sentence of paragraph should read "... we subsequently focused on understanding the relationship between these features in embryonal HB."

We have changed the text to reflect this comment.

Line 203: It is a stretch to call Axin2 and Sox4 as "markers" of embryonal histology. Rather they are upregulated in the embryonal regions as they are still expressed in the other regions.

We acknowledge this reviewer's comment that *AXIN2* and *SOX4* are expressed in both the fetal and embryonal components of hepatoblastoma and have removed instances in the text where we refer to *AXIN2* and *SOX4* as "markers" of embryonal histology. However, we note that we identified *AXIN2* and *SOX4* as more highly expressed in the embryonal components than in the fetal components, using both Smart-3SEQ and 10x Visium. Thus, although these genes are not embryonal markers per se, higher expression of *AXIN2* and *SOX4* in tumor regions can be used to distinguish embryonal from fetal histologies.

Legend for Fig. 2: Panel m is mislabelled as f

We thank the reviewer for noticing this error; we have revised Fig. 2.

Reviewer #2 (Remarks to the Author):

The authors have provided limited answer to the original comments. Some aspects are still problematic. The comments did raise a number of concerns which are confirmed by the new data provided. However, the authors have not considered these limitations in their data interpretation or their conclusions.

1. Figure 2b: it would be useful to show higher magnification of FGF19/Sox4 co-staining. Indeed, a large number of Sox4 positive cells seems to be located outside the FGF19 positive field.

Higher magnification image confirms that only a minority of cells co-expressing Sox4/FGF19. The authors need to change their conclusion accordingly.

We appreciate that this reviewer brings up a similar comment as reviewer 1. Indeed, only a subset of SOX4-positive cells are FGF19 expressing. We have now included more quantification of this data in Fig 2. Our conclusion is that FGF19 is expressed in a subset of SOX4 expressing cells, but those are the cells that have high nuclear beta-catenin. We have clarified this in our conclusions.

2. Figure 3c: The authors should generate a table showing the number of cells per cluster for each patient.

The table is provided figure 4?

Previously we had included a table with the number of cells for each patient. We have now added the number of cells per cluster per patient in Supplementary Data 4 (Excel file, Tab: Cell # per Cluster) as well as a panel to Fig. 4b (shown below) that indicates the percentage of cells in each cluster for each patient.

Fig. 4b:

3. The most interesting part of the manuscript is the derivation of Hepatoblastoma organoids. However, these organoids need to be better characterised to reach standards used in the field. It would be incredibly useful to show immunostaining for basic markers i.e ALB/AFP/AHNF4/KRT19 but also for specific markers such as Sox4 and MKi67. The authors should also show specific foetal/embryonic markers identified in their single cell analyses. They should also provide some sort of growth curve for few lines over 5 passages. Finally, they should analysed key markers over prolonged period of time to demonstrate phenotype stability (for example, ALB/AFP/HNF4 at passage 2, 5 and 10).

The authors did not provide immunostaining but in situ analyses. This is useful but do not fit the usual standard of characterisation. IF analyses on Alb/AFP/CK19 are very common in the field of liver organoids. They have to be able to provide such staining. Also, their comments on the lack of stability of their organoid need to be considered in their interpretation of their data and discussed in their conclusion. Again, the original comment did raise some concerns which is then confirmed by the authors but who chose to ignore it. Importantly, it seems that their culture selection ultimately selects for Chlangiocytes/biliary cells and they could lose the original hepatoblastoma. This could strongly affect their interpretation regarding the importance of FGF19. They need to show co-staining of CK19/AFP or CK19/ALB over a time course to determine the fraction of biliary/hepatoblastoma cells.

The reviewer mentions that the “organoids need to be better characterized to reach standards used in the field”. We would like to point out that:

- 1) While there may be standards for characterization of normal liver organoids, these are tumoroids, which may not conform to this reviewer’s assumptions about expression of normal liver markers.**
- 2) We are one of the first to grow tumoroids from hepatoblastomas and therefore are establishing the standards of this field.**
- 3) Just as cancer tissues are rarely histologically similar to each other (unlike normal tissues), tumoroids cannot be expected to be similarly organized (as normal organoids are).**

That said, we now include immunofluorescence of HNF4A, CK19, AFP, as well as the Wnt target gene TBX3, showing stability of marker expression in a non-FGF19 expressing tumoroid, HB1 (Fig. 3h). We also show the staining of these markers in early passage HB15 tumoroids, which contain FGF19-expressing cells that lack HNF4A staining (Fig. 3i). We further show that after multiple passages, HB15 loses the expression of HNF4A, suggesting that FGF19-expressing cells overtake the cultures (Fig. 3i). We have also included HNF4A/CK19 immunostaining of additional tumoroids from different patients in Supplementary Fig. S6b.

Fig. 3h-i:

The immunofluorescence data is consistent with the prior qRT-PCR data that we had included in our prior revision. We note, as we did previously, that because of the change in marker gene expression in HB15, and other FGF19-expressing tumoroids, we opted to perform all our functional experiments on early passage cells, where we verified that there were cells that still maintained HNF4A expression. We further show that these tumoroid cultures are not contaminated by normal biliary or hepatic cells, by overlaying the gene signatures of normal biliary and hepatic cells onto the scRNA sequencing data from the tumoroids (Fig. 4g, outlined in red dashes).

Fig. 4g:

Thus, we are not losing the original hepatoblastoma cells. Rather there is a specific population of hepatoblastoma cells, the FGF19-expressing cell population, that can become over-represented in the FGF19+ tumoroids (eg. HB15) at late passage. However, we show that in those tumoroids that never contained any FGF19-expressing cell population (eg. HB1), the hepatic marker genes are stable.

4. To define if the foetal / embryonic state corresponds to any developmental reality the authors should compare their hepatoblastoma organoids with hepatoblast organoid derived from foetal human liver (Wesley at al., Nature Cell Biology 2022). They could use the corresponding data to validate key markers.

They authors did the comparison but show very little data. It seems that the profile does not overlap

suggesting that their interpretation of foetal/embryonic signature is not accurate. Why not develop this aspect further? Again, the signature suggest a very strong biliary contamination which is likely to rapidly over grow the tumour cells.

Unfortunately, this comment appears to disregard the additional data and analyses that we had included in supplementary figures in the previous revision. We have now moved these analyses to main Fig. 4g (shown above, in response to comment 3) and describe the analyses in more detail in the text. We again note that these tumoroid cultures are not contaminated by normal biliary or hepatic cells, by overlaying the gene signatures of normal biliary and hepatic cells onto the scRNA sequencing data from the tumoroids in Fig. 4g (outlined in red dashes above, in response to comment 3). Rather, we have tumor cells that express a more biliary or more hepatic phenotype.

5. The approach used to define a foetal and embryonic signature is unclear. They need to provide the set of markers used to annotate the corresponding clusters. Again, the clusters currently idenGfied are not different cell type but different state of the same cells.

The approach used to define the fetal and embryonal gene signatures was shown in Fig. 1, using Smart-3SEQ histology-guided RNA sequencing to define genes that are differentially expressed in fetal vs. normal liver (fetal), embryonal vs. normal liver (embryonal), and embryonal vs. fetal (embryonal-specific or fetal-specific). We had previously included the lists of these marker genes in our supplementary data and refer the reviewer to these lists in Supplementary Table S2 and Supplemental Data 4. We analyzed our scRNA seq data using the AddModuleScore function in Seurat, using the above gene lists. We have previously detailed the Smart-3SEQ method in the text and have added clarifying text to indicate that we calculated the fetal or embryonal signature index using AddModuleScore.

See comment 4. They need to use previous publications on foetal tissue to annotate their clusters.

In the previous revision, we had included this information in the supplementary figures, which we believe this reviewer missed. We have now moved the data to main Fig. 4g (shown above in response to comment 3). We have discussed this data further in the main text.

6. It would be incredibly useful to perform RNA velocity and pseudotime analyses to define if there is any relation between the different cell clusters.

Again, they did not provide RNA velocity analyses but did perform pseudotime analyses. Pseudotime always requires a start point. However, the trajectory should be informative. If not, their interpretation of foetal/embryonic is not accurate. This could be a mix of cells without any link.

We have now included both RNA velocity and pseudotime analyses in Figure 4e. We ultimately assigned the starting point for the pseudotime plot based on what appeared to be the originating cells in the RNA velocity analysis. The analyses indicate that tumor cells expressing the more fetal-specific or embryonal-specific gene signatures derive from common progenitors expressing genes that are commonly upregulated in both the fetal and embryonal components as compared to normal liver.

Fig. 4e:

7. “Since the hepatoblastoma tumoroids grew in serum-free media containing well-defined chemical components and recombinant growth factors”. This statement is misleading. The organoids are grown in Matrigel which contains a diversity of factors (even the reduced version) and thus their media is far from defined. Furthermore, they use media containing FBS to passage their cells. The authors need to correct this statement and take this aspect in consideration.

Again, the authors should do an effort on this aspect. This is very simple. Matrigel contains growth factors which can influence their results and FBS is a problem. Why use FBS on the first instance? Organoids usually do not require FBS for passaging etc.

It is standard in organoid protocols to use 5% FBS in the washes to reduce attachment of organoids to plastic which leads to loss of cells/organoids. These procedures result in very miniscule amounts of total FBS, which the reviewer may not have realized. In fact, when we wash our organoids during passaging, we are left with organoids in <20 uL of wash media containing 5% FBS. This is then diluted in 1mL Matrigel (50x) and split among forty-eight 20uL Matrigel drops (50x), so the final concentration of FBS in each Matrigel drop is 0.002% at most. Further, each well of a 6-well plate contains eight 20ul Matrigel drops, which are then overlaid with 2ml serum-free media. Our data shows that without adding exogenous recombinant growth factors (either EGF+HGF+FGF10 or FGF19), the tiny amount of FBS and any growth factors within the Matrigel itself are not sufficient for either colony formation or tumoroid growth/proliferation. Thus, we can conclude that the residual FBS and Matrigel are not sufficient for tumor cell proliferation and can make conclusions from our experiments in which we add recombinant growth factors. We acknowledge the reviewer’s point that our growth conditions are not completely defined – however our experiments are testing specifically the role of exogenous recombinant growth factors that we are adding to the media. To the reviewer’s point, we have modified the text to state that the media is only *relatively* well defined. We have also added more details to the methods section on how the passaging of the tumoroids was performed.

8. The data on the importance of FGF19 are simply not convincing:

- “HB4, HB15, and HB17 showed a high percentage of cells expressing FGFs, particularly FGF19, by single cell RNA sequencing” (page 17). This statement does not fit the data provide in Sup. Fig. 3d. HB6 or HB7 seems to contain similar level of cells expressing FGFs?

- Similarly, the knock down of FGF19 does affect the clonality but this remains extremely small difference and the stat analysis seems problematic (Supplementary Fig. S5b).
- The data provided only demonstrate the importance on clonality and not proliferation. Organoids don't survive single cell dissociation and thus the data only show that FGF19 might have a role in survival in suboptimal culture conditions. To demonstrate effect on proliferation, the authors need to knock down FGF19 after organoid formation and then count the cells 10 days later.

These comments are really essentials. The authors answer them by excluding some of their original data to make them more consistent. Their justification is acceptable but again should impact the interpretation of their results. The growth curve is key. However, this experiment includes only 1 cell line with not statistical analyses (see figure 6d)?

We have revised Fig. 5e to include the data for all of the tumoroids in one graph, rather than separating the growth factor dependent and independent tumoroids. We have added Fig. 5f, which shows clearly that growth factor-dependent colony formation correlates with the percentage of FGF19-expressing cells in each tumoroid. Although HB6 and HB7 do contain some FGF19-expressing cells, the percentage is lower and they fall in the category of growth factor-dependent tumoroids when tested at low passage. We believe Fig. 5f strengthens our conclusions.

Regarding the tumoroid growth curve, we have now included additional repetitions and statistical analyses in Fig. 6d showing that in the absence of growth factors, knocking down FGF19 indeed results in a statistically significant reduction in growth that is partially rescued by exogenous FGF19.

9. MAPKinase pathway can be activated by a diversity of pathway especially insulin contains in the N2 used in their culture system. It would be super useful to provide Western blot analyses showing that addition of FGF19 does indeed increase the activity of this pathway.

The requested experiments were not performed. The justification on cell number is difficult to follow. You can grow organoid in optimal condition to obtain enough cells and then starve them from specific growth

factors for 1-2 hour (ERK phosphorylation change very rapidly) and then re-add specific combination of factors. In any case, their experiment show that other factors activate ERK and the authors should discuss this aspect in their conclusion.

We now include the experiment in which tumoroids are starved of growth factors and show that addition of FGF19 results in ERK phosphorylation, which is abrogated in the presence of Mek inhibitor or the complete absence of growth factors (Fig. 5d).

10. The molecular mechanisms seem to be only true for HB15 and thus can't be generalise. The author's answer does not fit the data provided. Again, a large number of functional analyses are only performed on HB15 based on their figure legend. Their results are sometimes very variable and are not reproducible across organoid lines which have nonetheless being carefully selected to fit their hypothesis. This could indeed be due to technical challenges and tumour variabilities. However, the authors must recognise these problems and tune down the generalisation of their conclusions.

We have previously performed the experiments depleting FGF19 in all three growth factor independent hepatoblastoma tumoroids (HB15, HB4, and HB17), showing its essentiality in colony formation. While in our previous revision, we showed the data for HB4 and HB17 in supplementary figures, we now include this in the main Fig. 6c.

We have further shown in both HB15 and HB17, that FGF19 depends on both beta-catenin and SOX4 (Fig. 7a, d, g). We also show that HB15 and HB17 have similar colony formation phenotypes when beta-catenin is inhibited (by DN TCF4) or SOX4 is knocked down (Fig. 7b-c, e-f, h-i). Based on these findings, we do believe that our conclusions are generalizable, however we have added a statement in our conclusion to address the limitations of our study given our small sample size.

Fig. 7a-i:

11. Sox4 knock down is extremely weak (and probably not statistically significant see Fig. 6d). Furthermore, the single cell analyses show that most cells expressing FGF19 do not express Sox4. So, the impact on FGF19 expression could be due to technical challenges associated with the knock down. Sox4 knock down is also likely to induce cell death (and thus the effect on FGF19 is indirect).

The knock down of Sox4 is still limited to 50% but statistical analyses are now provided. However, the exact statistical test used is not provided in the figure legend (Figure 7.d). How many experiments were performed? Which cell line? Which T-test? Can they really use paired comparison if those are independent experiments? , there is no error bar on the control? etc. They could have analysed cell death level after knock down and show decrease in CK19 expressing cells etc. Nonetheless, the authors did include this possibility in their revised conclusion.

Our single cell and in situ analyses have shown that almost all FGF19 expression occurs in a subset of cells that also express SOX4. We believe that our data in both HB15 and HB17 (Fig. 7d, g, above) is quite convincing that knocking down SOX4 results in downregulation of FGF19. Though the knockdown of SOX4 varies between experiments, we have further added a graph showing that the degree of FGF19 downregulation is directly proportional to the knockdown of SOX4 (Supplementary Fig. S11e).

Supplementary Fig. S11e:

Furthermore, we have now changed the display of our qPCR results with $\log_2(\text{fold change})$ rather than fold change and to demonstrate that by paired analyses of the cells expressing control and the test shRNA/DN TCF vector (Fig. 7, shown above). The statistical analyses are thus based on paired student t-test for each experiment in which the same cells are infected with control or shRNA.

Regarding cell death, the way the experiment is done is that the cells are transduced with lentiviral shRNA at hr 0, 12, 24, and 36, placed in antibiotic selection from hrs 48-120, and then either harvested for RNA analyses or plated for colony formation assays. Any early cell death between hrs 48-120 could be a combination of antibiotic selection and the effect of the knockdown, so we cannot accurately quantify cell death at the 120hr time point at which we analyzed RNA expression. We cannot rule out that this is already a selected population of surviving cells, however we do note that there is no decrease in KRT19 expression in the cells, suggesting that the SOX4-expressing/KRT19-expressing cells are not immediately dying. Thus, we extrapolate that the effects we are seeing are due to SOX4 depletion and not due to death of SOX4-expressing cells.

Reviewer #3 (Remarks to the Author):

The authors partly to my comments, modifications are challenging to identify since there is tracking and figures not numbered.

For ease of the reviewers, we have added the numbered figures with figure legends to the end of the main manuscript file. As with our previous revision, we added tracking to a separate file that we uploaded to the attachments, as there were so many modifications to the text that we felt it would be distracting in the main manuscript file.

1-Comparison with transcriptomic classifications previously published:

Utilizing recent RNAseq classification as a reference is essential and more precise than C1/C2. Inferring classification obtained from bulk transcriptomic in single-cell analysis is important, and most studies are performed in that direction. Organoids add complexity, but using classifications obtained in primary tumors as a reference is essential.

- the present single-cell results match more with the Hirsch et al. classification; it should be included in the principal figure in addition, or replacing, C1/C2.

- The lack of similarity with Nagae's classification is surprising it could be due to the choice of genes that are not well explored in single-cell data.

We thank this reviewer for their comments and believe they did not see the data added to supplementary figures in our previous revision. We have now moved all the transcriptomic classification to the main Figure 4f. Of note, we now also include analyses using recently published single cell multiomic analyses from Roehrig et al. 2024.

2-Functional analysis of FGF19 over-expression is still limited.

We have performed functional analyses of FGF19 to the best of our ability with the tumoroids. We have shown that FGF19 depletion in FGF19-expressing tumoroids reduces growth factor independent colony formation and growth, that FGF19 overexpression in FGF19-negative tumoroids is sufficient for growth factor independent growth.

REVIEWERS' COMMENTS

Reviewer #1 (Remarks to the Author):

I thank the authors for addressing this reviewer's concerns. Though some of the data that was requested during my first review was in fact present in their first revision, the presentation/readability has been significantly improved in this second version. Also, all the clarifying experiments have been performed.

Reviewer #2 (Remarks to the Author):

EDITORIAL NOTE: The response to this Reviewer's comments was assessed by Reviewer #1, who considered the concerns to be addressed.

Reviewer #3 (Remarks to the Author):

The authors answered to all my comments, I have no additional question.